# *Nomo1* deficiency causes autism-like behavior in zebrafish

Qi Zhang[1,5], Fei Li[1,5], Tingting Li[1], Jia Lin[1], Jing Jian[1], Yinglan Zhang[1], Xudong Chen[1], Ting Liu[1], Shenglan Gou[1], Yawen Zhang[1], Xiuyun Liu[1], Yongxia Ji [1], Xu Wang[2,3,4] & Qiang Li [1]✉

## Abstract

**Patients with neuropsychiatric disorders often exhibit a combination of clinical symptoms such as autism, epilepsy, or schizophrenia, complicating diagnosis and development of therapeutic strategies. Functional studies of novel genes associated with co-morbidities can provide clues to understand the pathogenic mechanisms and interventions. *NOMO1* is one of the candidate genes located at 16p13.11, a hotspot of neuropsychiatric diseases. Here, we generate *nomo1*−/− zebrafish to get further insight into the function of NOMO1. *Nomo1* mutants show abnormal brain and neuronal development and activation of apoptosis and inflammation-related pathways in the brain. Adult *Nomo1*-deficient zebrafish exhibit multiple neuropsychiatric behaviors such as hyperactive locomotor activity, social deficits, and repetitive stereotypic behaviors. The Habenular nucleus and the pineal gland in the telencephalon are affected, and the melatonin level of nomo1−/− is reduced. Melatonin treatment restores locomotor activity, reduces repetitive stereotypic behaviors, and rescues the non-infectious brain inflammatory responses caused by *nomo1* deficiency. These results suggest melatonin supplementation as a potential therapeutic regimen for neuropsychiatric disorders caused by *NOMO1* deficiency.**

**Keywords** *Nomo1*; Neuropsychiatric Disorders; Serotonin; Inflammatory Response; Melatonin
**Subject Category** Neuroscience

## Introduction

Neurodevelopmental and psychiatric disorders including autism, schizophrenia, epilepsy, intellectual disability, and so on. These disorders have complex genetic etiologies and exhibit a variety of abnormal behavioral traits such as anxiety, impaired social interaction, and repetitive stereotypic behaviors, and there is a lack of effective medications to treat these symptoms (Sahin and Sur, 2015; Willsey et al, 2018). NOMO1 is located at 16p13.11, multiple genetic analyses of large samples have shown that chromosomal segment 16p13.11 is strongly associated with neuropsychiatric disorders (Ramalingam et al, 2011; Ullmann et al, 2007; de Kovel et al, 2010), suggesting a potential role for NOMO1 in neuropsychiatric disorders, particularly autism and epilepsy. In 2010, Williams et al performed a genome-wide analysis of 410 children and showed that duplicated copy number variants of NOMO1 were associated with ADHD (Williams et al, 2010). In 2015, Tassano et al found deletion-type copy number variants in NOMO1 associated with autism, intellectual disability, and epilepsy (Tassano et al, 2015). Brownstein et al showed that NOMO1 variants associated with epilepsy and autism (Brownstein et al, 2016).

The contributing factors explaining how *NOMO1* deficiency results in neuropsychiatric disorders remain unclear. *NOMO1* is a negative regulator of the Nodal signaling pathway, while Nodals are essential for the formation of the neuroectoderm and mesoderm during the development of early embryos. NOMO1, TMEM147, and nicalin form protein complexes that inhibit the nodal signaling pathway during the early development of zebrafish (Haffner et al, 2007; Dettmer et al, 2010). *NOMO1* is a candidate gene associated with glioma, early-onset colorectal cancer and facial asymmetry (Cao et al, 2018; Huang et al, 2017; Perea et al, 2017), and *nomo1* mutation in zebrafish resulted in chondrodysplasia (Cao et al, 2018). By determining the expression pattern of *nomo1* in zebrafish using whole-mount in situ hybridization (WISH), the *nomo1* gene was shown to be expressed at high levels in the anterior mesendoderm and endoderm during early embryonic development and was abundant in the brain of larval zebrafish (Haffner et al, 2004).

Neurotransmitters have an important role in brain development and functional circuitry, and a variety of neuropsychiatric disorders exhibit varying degrees and types of abnormal neurotransmitter levels. For instance, elevated whole blood serotonin (also called 5-HT) was the first biomarker identified in autism, and the phenomenon is present in more than 25% of affected children (Muller et al, 2016). Serotonin levels are also significantly elevated in the serum of patients with epilepsy following generalized seizures

[1]Translational Medical Center for Development and Disease, Shanghai Key Laboratory of Birth Defect Prevention and Control, NHC Key Laboratory of Neonatal Diseases, Institute of Pediatrics, Children's Hospital of Fudan University, National Children's Medical Center, 210013 Shanghai, China. [2]Cancer Institute, Pancreatic Cancer Institute, Fudan University Shanghai Cancer Center, 200032 Shanghai, China. [3]Shanghai Pancreatic Cancer Institute, Shanghai Key Laboratory of Radiation Oncology, Fudan University Shanghai Cancer Center, Fudan University, 200032 Shanghai, China. [4]Key Laboratory of Metabolism and Molecular Medicine, Ministry of Education, School of Basic Medical Sciences, Fudan University, 200032 Shanghai, China. [5]These authors contributed equally: Qi Zhang, Fei Li. ✉E-mail: liq@fudan.edu.cn

(Akyuz et al, 2021). Genetic variants in the serotonin pathway have also been strongly associated with schizophrenia onset (Hrovatin et al, 2020). In addition, abnormalities in glutamate, aminobutyric acid, and noradrenergic neurotransmitters were also reported in cases of clinical neuropsychiatric disorders (Chao et al, 2010; O'Donovan et al, 2017).

Here, we generated an in vivo *nomo1* KO model zebrafish to explore the underlying functional mechanisms. *nomo1* deficiency in zebrafish causes abnormal development of the central nervous system, and multiple neurons were affected as development proceeds. The adult brain of $nomo1^{-/-}$ exhibited significantly less mass, fragile tissues, and increased apoptosis. Meanwhile, the *nomo1* mutant zebrafish exhibit anxiety symptoms such as overactive locomotor activities, and autism-like phenotypes such as repetitive behaviors and social defects. In addition, through studies on how *nomo1* deficiency conspicuously influences brain development and neurotransmitter metabolism, we reported inhibited serotonin converted to melatonin pathway is one of the important reasons why *nomo1* exhibits abnormal behavioral characteristics, and melatonin treatment significantly rescued the overactive locomotor activities and repetitive behaviors of $nomo1^{-/-}$ zebrafish.

# Results

## The mRNA expression levels of *nomo1* from early embryonic development to the juvenile stage and generation of *nomo1* mutant zebrafish

Zebrafish *nomo1* is homologous to human *NOMO1*, with 68% and 70% homology in cDNA and amino sequence, respectively. Cao et al and Haffner et al reported the expression pattern of *nomo1* in zebrafish from early embryonic development to larval stage using WISH (Haffner et al, 2004; Cao et al, 2018). In order to investigate the expression level of Nomo1 in different developmental stages, we performed RT-qPCR analysis. From 12 hpf to 7 dpf, the Nomo1 expression level was detected at whole embryo and larvae levels, while at 14 dpf, 1 mpf and 2 mpf, it was detected at whole brain levels. Results showed that the expression of *nomo1* mRNA increased before 48 hpf. As development progressed, a second peak of *nomo1* mRNA expression appeared at 14dpf, and the highest mRNA expression level was detected in brain tissues from 2-mpf zebrafish (Fig. 1A).

Then, we generated *nomo1* knockout zebrafish using the CRISPR/Cas9 system to investigate the function of it. The sgRNA of *nomo1* was designed to target exon 7, which was located before the functional structural domain EMC7-beta-sandw (analyzed from pfam database, numbered PF09430). EMC7-beta-sandw is a beta-sandwich domain found in ER membrane protein complex subunit 7(EMC7), which is an integral membrane component of the EMC. EMC mediate the insertion of newly synthesized membrane proteins into endoplasmic reticulum membranes. EMC7-beta-sandw domain may be associated with the embedding of Nomo1 in the ER membrane. Figure 1B shows diagram of sgRNA target and 1-base deletion resulted truncated protein, which was early terminated after exon 7 translation. And Fig. EV1 shows the amino acid sequence of wild type (1220 amino acids) and mutant Nomo1 (232 amino acids). Genotyping sequence of mutant *nomo1* were shown in Appendix Fig. S1. RT-qPCR analysis confirmed that

*nomo1* mRNA expression was significantly reduced at three expression peak-time points in $nomo1^{-/-}$ zebrafish (Fig. 1C).

## Morphological analysis of $nomo1^{-/-}$ zebrafish revealed abnormalities in early development and adult brain

To investigate the effect of *nomo1* knockout on zebrafish development, we performed morphological analysis first. A low percentage of mutant embryos exhibited morphological abnormalities such as delayed development including prolonged rupture of fetal membranes, tail bending, yolk malformation including bigger yolk, smaller yolk and yolk extension malformation and pericardial edema (Fig. EV2A,B,G). However, these morphological abnormalities of $nomo1^{-/-}$ zebrafish gradually became less noticeable during development (Fig. EV2C–E), and the survival rate and body length of zebrafish at different developmental stages did not exhibit significant differences (Fig. EV2F,I). Cao et al showed the zebrafish *nomo1* homozygous mutants they generated exhibited chondrodysplasia and skeletal development defects, yet we did not observe similar results (Fig. EV2H).

Next, we performed morphological structure analysis of the adult zebrafish brain. The $nomo1^{-/-}$ brain did not show significant size differences (Appendix Fig. S2), however, mass of $nomo1^{-/-}$ brain was significantly lower than wild type (Fig. 1D). Further, brain tissue section analysis revealed that $nomo1^{-/-}$ brain was more fragile and highly susceptible to disruption of tissue integrity by mechanical forces of sectioning. In addition, significantly reduced neuronal cells were identified in several nucleus, including VP, PM and PPp of forebrain and Ha, pineal gland of the telencephalon. Vam, VamG nucleus of the midbrain showed a decreasing trend (Fig. 1E,F).

## *Nomo1* deficiency significantly affected the locomotion of 7-dpf larval zebrafish

Given that neuropsychiatric disorders associated with *nomo1* deletion mutations are characterized by a variety of abnormal motor behaviors, we performed a series of motor and behavioral analyses. Firstly, we examined the locomotor activity of 7-dpf zebrafish and their reactions to light–dark transitions to determine whether the loss-of-function of *nomo1* would modulate the behavior of larval zebrafish. After 20 min of adaptation, the locomotion and thigmotaxis behaviors of larval zebrafish were analyzed (Fig. 2A). We recorded the tracks of the swimming behaviors of WT and mutant zebrafish in 24-well plates (Fig. 2B). An analysis of the average distance moved per minute from minutes 21 to 60 (L0) under light conditions (Fig. 2C) showed that the locomotion of $nomo1^{-/-}$ zebrafish was significantly reduced compared with $nomo1^{+/+}$ zebrafish.

Similarly, the mutant zebrafish displayed significantly decreased locomotor activity during the two light–dark cycles. And both WT and mutant zebrafish showed a light-sensitive reaction in every light–dark cycle (Fig. 2D,E). Figure 2F shows the ratio of the average distance in dark to the average distance in light. The $nomo1^{-/-}$ zebrafish showed a significantly more intense response to lighting changes in the first light–dark cycle than WT zebrafish, suggesting they were more easily disturbed by environmental changes (Fig. 2F). In the second light–dark cycle, there was no significant difference in the degree of response between the two

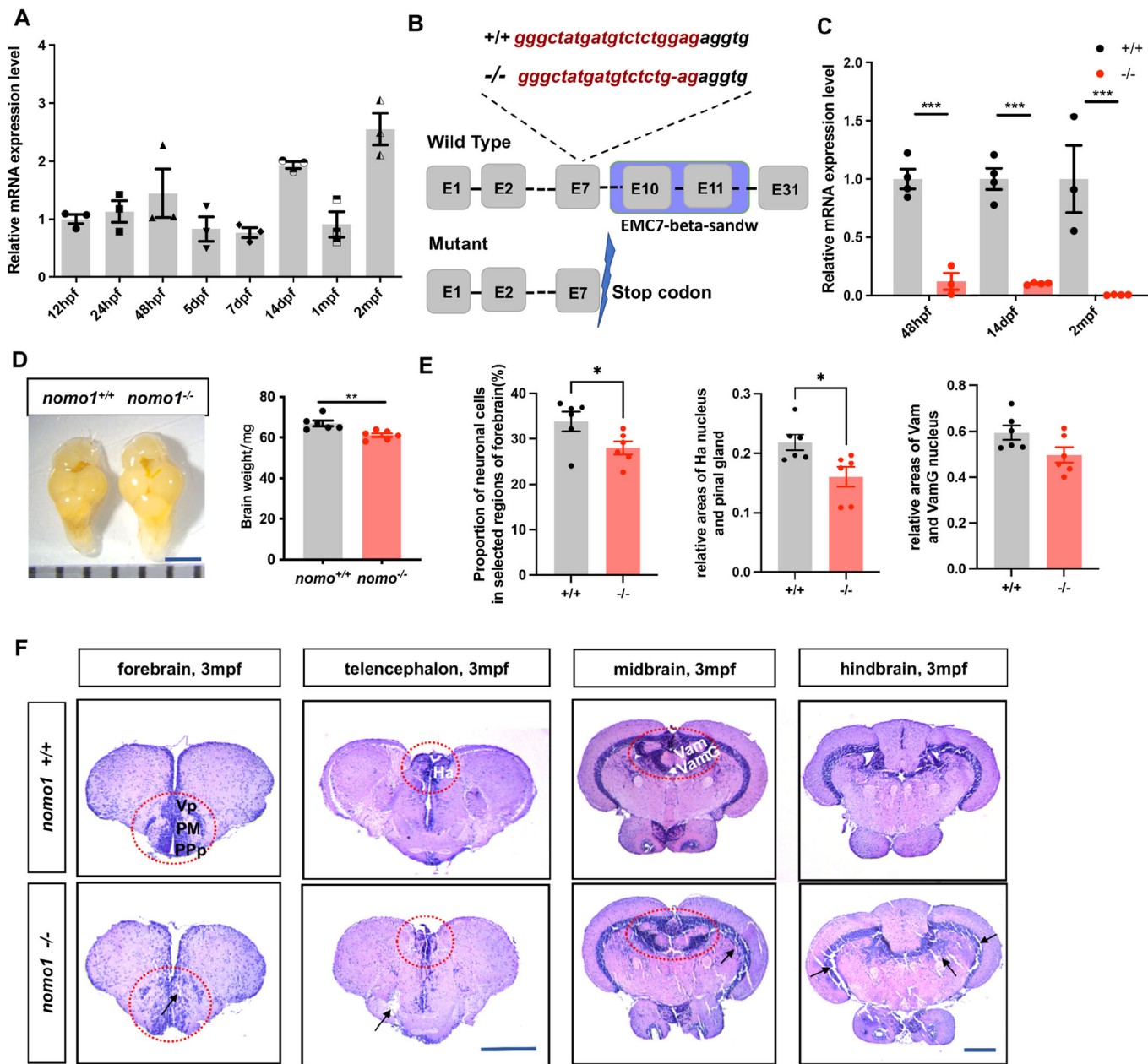

**Figure 1. The expression level of the *nomo1* mRNA during development and generation of *nomo1* mutant zebrafish.**

(A) Relative *nomo1* transcript levels at different developmental stages (biological replicates, $n = 3$). (B) Diagram of zebrafish *nomo1* gene and mutation induced by CRISPR/Cas9, gray boxes indicate exons. Target site was on Exon 7 resulting in a 1-base deletion (shown in sequence). The mutation results in early termination of translation after exon 7, before the functional domain of Nomo1, EMC7-beta-sandw. (C) Reduced expression of *nomo1* mRNA in the whole embryo at 48 hpf, heads at 14 dpf and brain tissues at 2 mpf of *nomo1^{-/-}* zebrafish analyzed using RT-qPCR (biological replicates, $n = 4$). (D) Brain weight of *nomo1^{+/+}* and *nomo1^{-/-}* shows significant differences (biological replicates, $n = 6$, 20 brains for each replicate group). Scale bar $= 1$ mm. (E) Quantitative analysis of brain cells or tissues (biological replicates, $n = 6$). (F) Brains of *nomo1^{-/-}* exhibit relatively fragile brain tissue, arrows indicate tissue that collapses immediately after sectioning due to being too fragile. Red dashed circles indicate nucleus with abnormal neuronal development. Scale bar $= 25$ µm. Data information: Data are analyzed using unpaired *t* test and presented as the means ± SEM. *$P < 0.05$, **$P < 0.01$, ***$P < 0.001$, ****$P < 0.0001$. Source data are available online for this figure.

strains, indicates *nomo1^{-/-}* zebrafish adapted the light–dark change (Fig. 2F).

In addition, we measured the thigmotaxis of WT and mutant zebrafish under light conditions by recording the percentage of time spent/distance moved in the inner zone (Fig. 2G). Thigmotaxis

is another validated index of anxiety. Animals that are engaged in thigmotaxic behavior tend to move in close proximity to the boundaries of the environment (Schnörr et al, 2012). The nomo1 mutant zebrafish exhibited increased thigmotaxis behavior, indicating edge preference under continuous illumination (Fig. 2H).

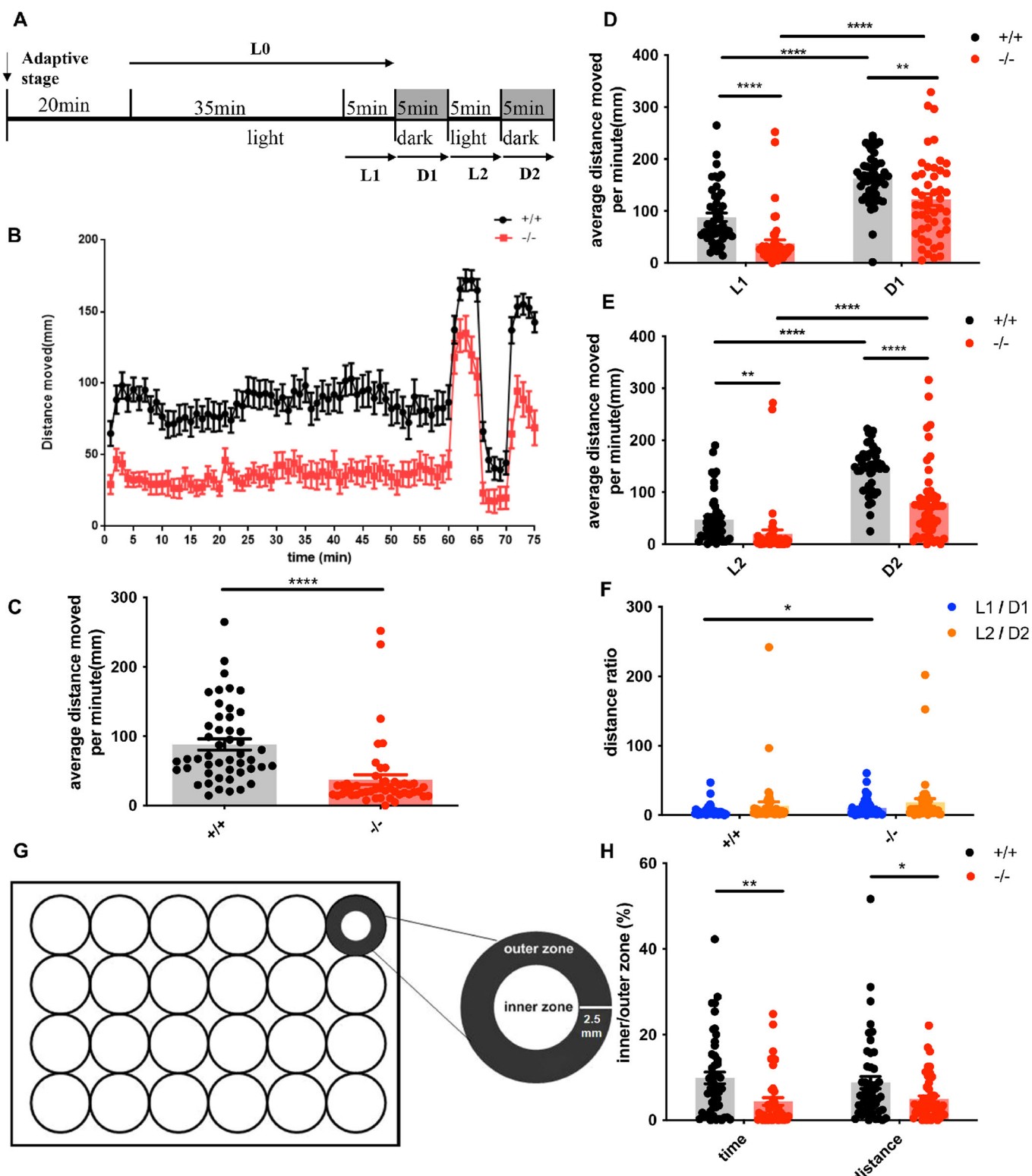

## The 15-dpf, 1-mpf, and 3-mpf *nomo1⁻/⁻* zebrafish showed increased locomotion

Though locomotor behavior was decreased of larva *nomo1⁻/⁻* zebrafish (7 dpf), this result was instead reversed during development. Depending on the body size of the zebrafish, the size of the container for behavioral analysis experiments changed (Fig. 3A,C). Diagram of Fig. 3B,D,E shows the swimming trend of WT and *nomo1⁻/⁻* zebrafish during the experimental period. The average distance moved per minute of *nomo1⁻/⁻* zebrafish was significantly enhanced than WT (Fig. 3F).

**Figure 2.   The locomotion and thigmotaxis behavior of 7-dpf WT and mutant zebrafish.**

(A) Process of light/dark test of larval *nomo1*$^{+/+}$ and *nomo1*$^{-/-}$ zebrafish at 7 dpf. The whole experiment lasted for 75 min and consisted of 40 min of light (L0) and two 5-min light/dark cycles (L1-D1 and L2-D2). (B) Trend of total distance swum by larval zebrafish during the experiment. The horizontal axis is the swimming time, and the vertical axis is the distance moved to visualize the locomotion of zebrafish during the experiment (biological replicates, $N = 48$). (C) The average distance moved per minute under continuous illumination is shown (biological replicates, $N = 48$ per genotype). (D, E) The average distance moved per minute under two light/dark cycles (biological replicates, $N = 48$ per genotype). (F) Ratio of the distance moved by larval zebrafish during each light/dark cycle (biological replicates, $N = 48$ per genotype). (G) The inner and outer zones were delineated as shown in 24-well plate. The ratio of distance moved/time spent in the inner zone were to measure the frequency of thigmotaxis behavior. (H) The ratio of distance moved/time spent in the inner zone under light conditions (L0 period) in 7-dpf zebrafish larvae ($N = 48$). Data information: Data are analyzed using unpaired $t$ test and presented as the means ± SEM, $^*P < 0.05$, $^{**}P < 0.01$, $^{****}P < 0.0001$. Source data are available online for this figure.

## *nomo1*$^{-/-}$ zebrafish exhibited social behavior disorders and repetitive stereotype behavior

Zebrafish are gregarious animals and have a strong tendency to be socially active. The social preference behavior (Movies EV1 and EV2) and shoaling behavior (Movies EV3 and EV4) are indicators of zebrafish social tendencies. Here, we analyzed the social behavior performance of both strains. In social preference test, *nomo1*$^{+/+}$ zebrafish swam along the social area throughout the experiment, whereas the *nomo1*$^{-/-}$ zebrafish swam in a dispersed and random manner. The heatmaps in Fig. 4B,C were obtained by analyzing the trajectory of the zebrafish in the mating tank. *nomo1*$^{-/-}$ zebrafish spent significantly less time and moved significantly less distance in the social area than *nomo1*$^{+/+}$ zebrafish (Fig. 4D,E).

The shoaling behavior was detected to assess the social skills of WT and *nomo1*$^{-/-}$ zebrafish. Generally, *nomo1*$^{+/+}$ zebrafish swam together in an open-field test, reflecting the social nature of the species. Compared to *nomo1*$^{+/+}$ zebrafish, *nomo1*$^{-/-}$ zebrafish exhibited increased interindividual distance from their companions, indicating that the *nomo1*$^{-/-}$ zebrafish are less capable of gregariousness (Fig. 4G).

Repetitive behaviors are one of the distinguishing behavioral characteristics of individuals with autism (Lai et al, 2014), here we detect and measure three kinds of repetitive behaviors in zebrafish include back-and-forth motion, small circling, and big circling movement. The trajectories of the repetitive behaviors are shown in Fig. 4H, detailed information of these repetitive behaviors, see Movies EV5–EV7. Results analyzed from swimming movement of *nomo1*$^{-/-}$ zebrafish showed that mutant zebrafish exhibited significantly increased repetitive stereotype behaviors (Fig. 4I).

## *nomo1 deficiency* affects multiple neuronal development

During early embryonic development, as we have shown above, *nomo1* expression peaks at 24 to 48 h, which is also a critical period for brain primordium development. To explore the molecular and cellular mechanisms underneath *nomo1* deficiency, we performed qRT-pcr to analysis expression levels of a series of brain and neuron development-associated genes. Results showed that neuro progenitor marker *neurog1* and *huc* (Hoijman et al, 2017; Kim et al, 1997)were downregulated. Motor neuron marker *islet* (Baeuml et al, 2019), which has additional ventral expression domains extending into midbrain were inhibited. hindbrain marker *egr2b* and ectoderm marker *foxb1a*, which was then expressed in the midbrain and hindbrain (Li et al, 2020), were also inhibited (Fig. EV3A). Meanwhile, the mRNA expression levels of forebrain marker *fezf2* (Shimizu and Hibi, 2009), dopaminergic neurons

marker *sim1a* and *th* (Schweitzer et al, 2013), GABAergenic neuron marker *gad1b* (Zhang et al, 2020) did not differ significantly but showed a decreasing trend (Fig. EV3C).

Whole-mount in situ hybridization analysis also displayed inhibited expression of *neurog1*, *foxb1a* and *islet* in midbrain and hindbrain (Fig. EV3B) and a decreasing expression of *fezf2*, *sim1a*, *gad1b*, and *th* (Fig. EV3D).

To directly observe the development of distinct neuronal populations, we employed three types of transgenic zebrafish lines: one featuring enhanced green fluorescent protein (EGFP) labeling GABAergic inhibitory neurons, another expressing dsRed labeling glutamatergic excitatory neurons, and a third displaying red fluorescent protein (RFP) labeling neuronal precursor cells. Results showed that at 48 hpf and 3 dpf, *nomo1*$^{-/-}$ exhibited varying degrees of reduction in GABAergic inhibitory neurons and glutamatergic excitatory neurons across the forebrain, mesencephalon, midbrain, hindbrain, and spinal cord when compared to wild-type zebrafish (Fig. 5A,B,E). In addition, at 3 dpf and 6 dpf, *nomo1*$^{-/-}$ displayed a decreased abundance of neuronal precursor cells compared to the wild type (Fig. 5C,D,F). These findings collectively indicate that the development of multiple neuronal populations is compromised in *nomo1*$^{-/-}$.

Furthermore, our analysis of the adult brain revealed a significant reduction in the expression levels of multiple neuronal markers (Fig. 5G). In addition, the expression of the apoptosis-promoting gene *bbc3* was markedly upregulated, while the apoptosis-inhibiting gene *bcl21l* was significantly downregulated (Fig. 5H). Although the expression of the apoptosis-promoting gene *pidd1* was notably decreased, and the apoptosis-inhibiting gene *bcl2a* did not exhibit significant differences. These findings suggest an increase in apoptotic processes within the adult brain of *nomo1*$^{-/-}$.

The results indicate that *nomo1* regulates early neural developmental patterning in vivo. Since the midbrain and hindbrain are closely associated with motor control, motor coordination and procedural memory, this further suggests a physiological basis for the abnormal behavioral phenotype of *nomo1*$^{-/-}$.

## Transcriptome analysis of *nomo1* knockout zebrafish

To further explore the molecular mechanism beneath the *nomo1* deficiency resulted autism-like phenotypes, we performed transcriptome sequencing analysis to identify differentially expressed genes and enriched pathways.

A volcano map visually shows the distributions of DEGs, 292 genes were downregulated and 254 genes were upregulated at the transcriptional level in mutant zebrafish (Appendix Fig. S3). The altered DEGs are listed, and their $P$ values are shown in Table EV2.

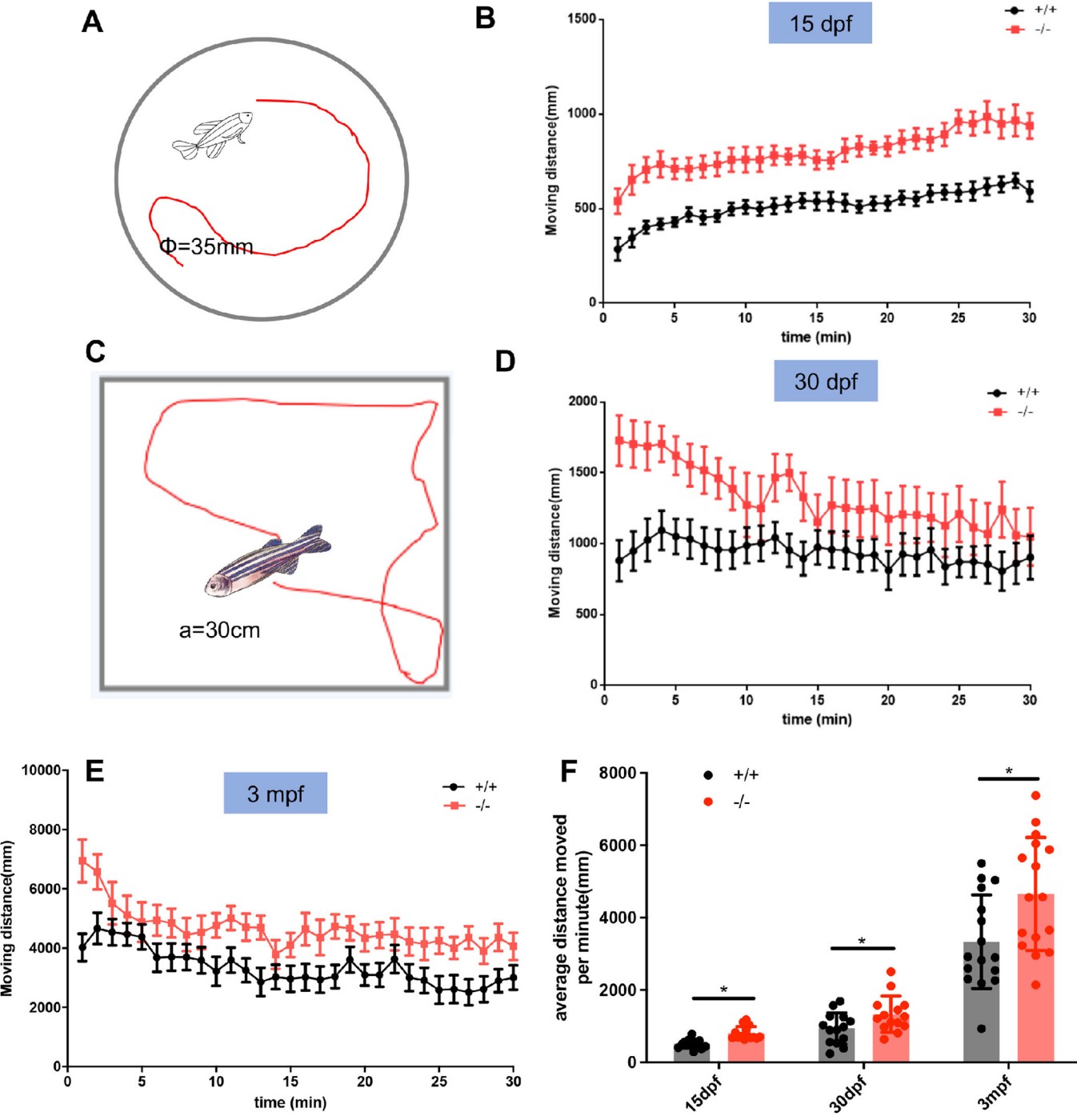

**Figure 3. The locomotion of larval, juvenile, and adult WT and nomo1⁻/⁻ zebrafish.**

(**A**) Container of 15-dpf zebrafish in the open-field experiment. (**B–E**) (**C**) Container of 30 dpf and 3-mpf zebrafish in the open-field experiment. (**B, D, E**) The locomotion of WT and mutant zebrafish over the total 30-min experimental period (biological replicates, $N = 16$ for each genotype at different developmental stages). (**F**) The average distance moved within each 1-min bin under continuous illumination is plotted (biological replicates, $N = 16$). Data information: Data are analyzed using unpaired $t$ test and presented as the means ± SEM, $^*P < 0.05$. Source data are available online for this figure.

We then performed GO annotation and KEGG enrichment pathway analyses to excavate the functional relationships between 556 DEGs. Figure 6A shows top 20 significantly enriched GO pathway, immune-related biological processes including antigen processing and presentation, classical pathway complement activation and immunoglobin-mediated immune response showed significant differences.

KEGG enrichment analysis revealed that there are many other affected pathways besides the immune-related ones. These pathways are mainly involved in organic metabolism and fatty acid biosynthesis. Although

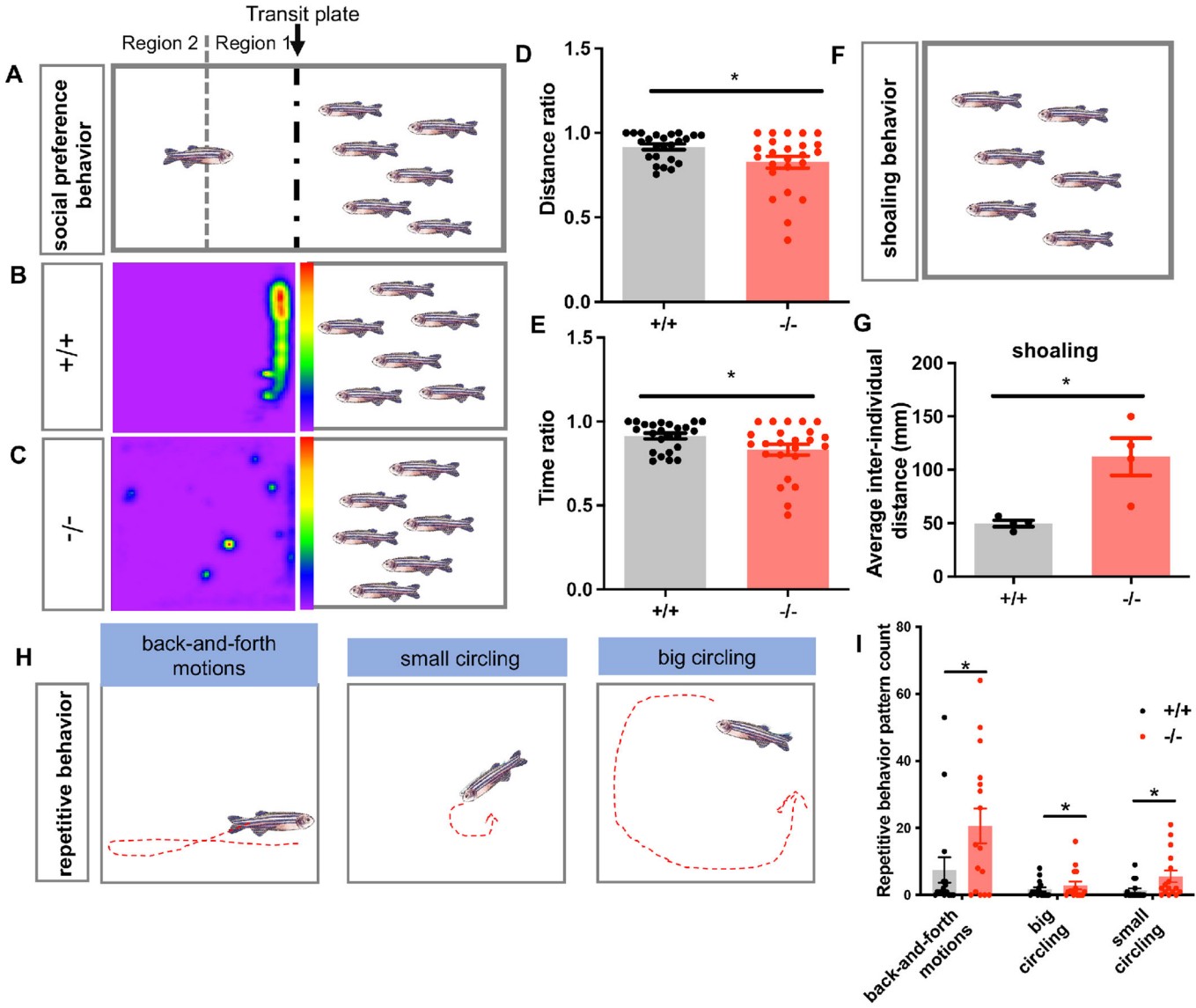

**Figure 4.** *nomo1⁻ᐟ⁻* zebrafish displayed social deficiency and repetitive behaviors.

(A–E) The social preference experiment with 2-mpf zebrafish. (A) Schematic diagram of the individual social behavior experiment. A heatmap (B, C) showing that *nomo1⁻ᐟ⁻* zebrafish spent significantly less time in the social area than WT zebrafish. The ratio of distance from the social area (D) and time spent in the social area (E) was significantly lower for *nomo1⁻ᐟ⁻* zebrafish than for WT zebrafish (biological replicates, $N = 20$). (F, G) Schematic of the shoaling test in which the interindividual distance exhibited by *nomo1⁻ᐟ⁻* zebrafish was significantly higher than WT zebrafish (biological replicates, $N = 4$). (H, I) Schematic of different repetitive behaviors including "back-and-forth motions", "small circling" and "big circling". *nomo1⁻ᐟ⁻* zebrafish exhibited a significantly higher proportion of all kinds of repetitive behaviors (biological replicates, $N = 16$). Data information: Data are analyzed using unpaired *t* test and presented as the mean ± SEM, $^*P < 0.05$. Source data are available online for this figure.

there were no significant differences in the degree of enrichment, these pathways are listed in Fig. 6B due to some of their key genes exhibit differential expression (see Table EV3 for affected key genes).

## Minocycline partially inhibits brain inflammatory response and reduces small circling repetitive behavior in *nomo1⁻ᐟ⁻*

Transcriptome analysis suggested that *nomo1* deletion mutations lead to abnormalities in immune-related pathways in the brain. To verify whether *nomo1⁻ᐟ⁻* brain exhibits inflammatory responses, we examined

the expression of inflammatory cytokines such as IL1β, TNFα, and IL6. The results revealed that the expression of them were significantly increased (Fig. 6C). Our previous work showed minocycline (MC) treatment inhibit the inflammatory response in the brain of genetically defective autism model zebrafish, and significantly rescued the social behavior abnormalities (Zhang et al, 2022). We therefore treated *nomo1⁻ᐟ⁻* with MC and examined whether its behavioral characteristics could be restored to normal. The results showed that after MC treatment, the expression of IL1b was significantly suppressed, but the expression of IL6 was significantly increased (Fig. 6C). Locomotor activity (Fig. 6D), and repetitive stereotypic behaviors such as back-and-forth swimming and

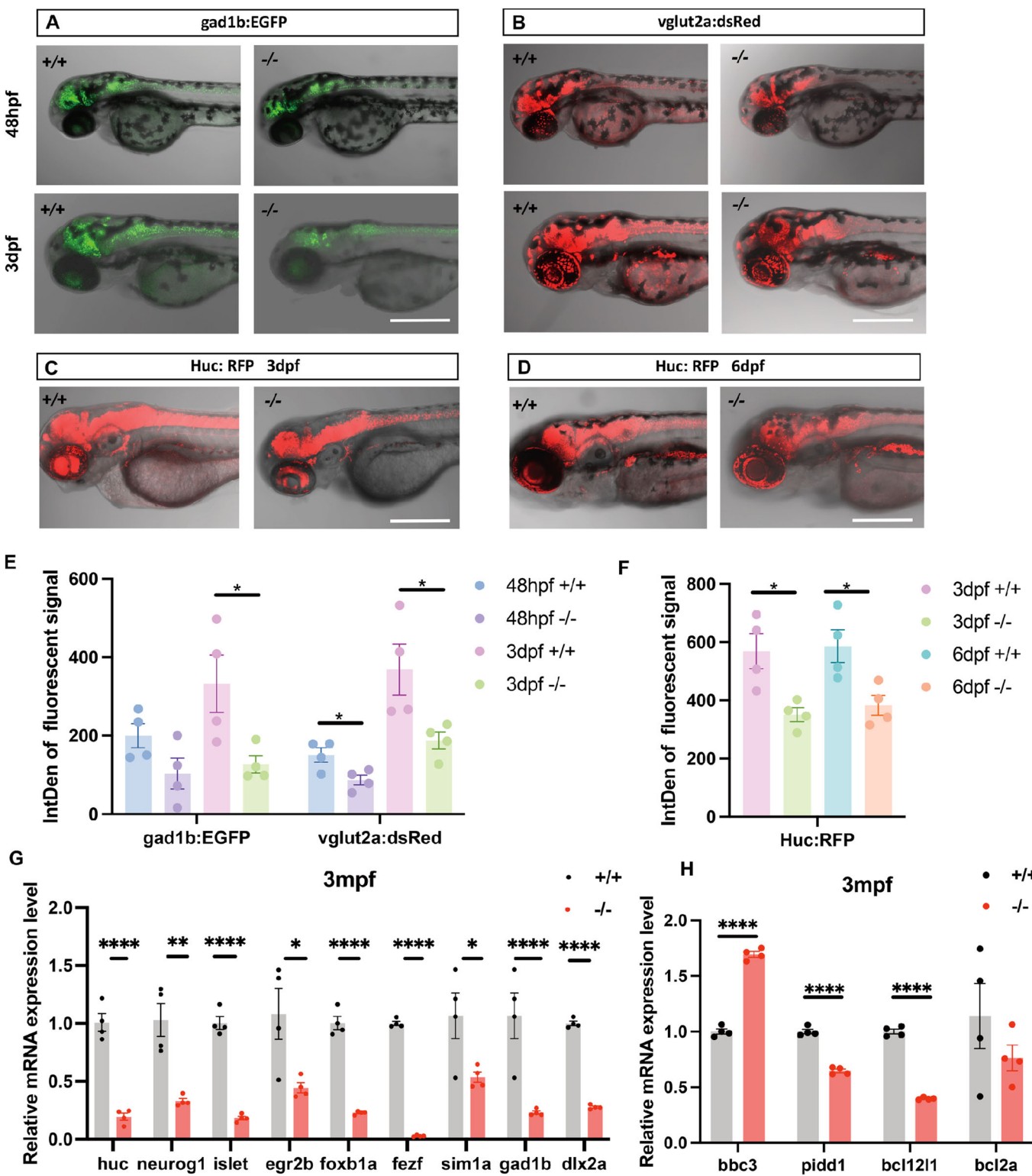

large circle swimming remained at abnormally high levels (Fig. 6F), and social deficits were not rescued (Fig. 6E; Appendix Fig. S4). However, repetitive stereotypic behaviors of small-circle swimming were restored to normal levels (Fig. 6F). These results suggest that MC treatment partially suppressed the inflammatory brain response in *nomo1*^−/− and partially rescued its repetitive stereotypic behaviors.

## Expression levels of neurotransmitters and metabolites were altered in the brains of *nomo1*^−/− zebrafish

Levels of neurotransmitters reflect neuronal signal communication in the brain. We further examined neurotransmitters and metabolites in zebrafish brains to reveal the mechanism underlying

**Figure 5. Loss-of-function of *nomo1* affected neurodevelopment in larval zebrafish.**

(A, B) EGFP labeled GABAergic inhibitory neurons and dsRed labeled glutamatergic excitatory neurons at 48 hpf and 3 dpf of WT and *nomo1*⁻/⁻. Scale bar=1 mm. (C, D) RFP labeled neuro progenitors at 3 dpf and 6 dpf of WT and *nomo1*⁻/⁻. Scale bar=1 mm. (E, F) Quantitative analysis of fluorescent signal (biological replicates, $n = 4$). (G) The relative expression level of neurological genes in *nomo1*⁻/⁻ brain at 3 mpf ($N = 4 \times 5$ animals per group). (H) The relative expression level of apoptosis genes in *nomo1*⁻/⁻ brain at 3 mpf ($N = 4 \times 5$ animals per group). Data information: Data are analyzed using unpaired $t$ test and presented as the means ± SEM. $^*P < 0.05$ and $^{**}P < 0.01$, $^{****}P < 0.0001$. Source data are available online for this figure.

autism-like behaviors. We analyzed the following 13 neurotransmitters and their metabolites using LC-MS/MS: $r$-amino-butyric acid, levodopa, DA, epinephrine, 5-HIAA, serotonin, 3-methoxytyramine, acetylcholine, histamine, normetanephrine, tyramine, glutamate, and glutamine. Appendix Fig. S5A shows the cluster relationships of each metabolite between *nomo1*⁺/⁺ and *nomo1*⁻/⁻ zebrafish, and the tree structure at the left shows the clustering relationship of each group. The results of the hierarchical clustering analysis showed increased levels of ten neurotransmitters and their metabolites in *nomo1*⁻/⁻ zebrafish compared with *nomo1*⁺/⁺ zebrafish. Detailed descriptions of the relative levels of the different metabolites are shown in Fig. 7A, the expression of γ-aminobutyric acid, levodopa, epinephrine, serotonin, 3-methoxytyramine, histamine, normetanephrine, tyramine, glutamate, and glutamine were significantly higher in *nomo1*⁻/⁻ zebrafish than in *nomo1*⁺/⁺ zebrafish.

The above results were validated using HPLC analysis, and the same trend results were obtained, the levels of serotonin and norepinephrine were significantly increased (Fig. 7B).

### Abnormal serotonin metabolism leads to reduced melatonin levels and melatonin treatment rescues overactive locomotor behavior and repetitive stereotypic behavior in *nomo1*⁻/⁻

The abnormal levels of neurotransmitters and their metabolites provided us with new clues to analyze the molecular mechanisms underlying the pathogenesis of *nomo1* deletion mutations. After cross-referencing the anabolic pathways of the abnormal neurotransmitters with the DEGs analyzed from transcriptome analysis, results showed that there were no significant differences in the expression levels of a series of genes associated with serotonin synthesis, transport, and reuptake (Table EV4), whereas the abnormal expression levels of *asmt* during tryptophan metabolism may underlie the abnormally elevated serotonin levels (Appendix Fig. S6). LC-MS/MS analysis confirmed a significant decrease in melatonin levels in the brain of *nomo1*⁻/⁻ compared to the wild type (Fig. 7C).

Previous research showed reduced melatonin levels lead to anxious behavior in animals (Fenton-Navarro et al, 2021; Wang et al, 2021), and we tested whether melatonin treatment of *nomo1*⁻/⁻ has an ameliorative effect on their behavior. The results showed that MT treatment significantly rescued the overactive locomotor behavior of *nomo1*⁻/⁻ (Fig. 7D), and a series of repetitive stereotypic behaviors, including back-and-forth swimming, and swimming in large and small circles were also significantly rescued (Fig. 7F). However, MT treatment did not rescue socially deficient behaviors of *nomo1*⁻/⁻ (Fig. 7E; Appendix Fig. S7).

### Inhibition of serotonin rescues overactive locomotor behavior, and exacerbated social deficits in *nomo1*⁻/⁻

Serotonin can also play a role in anxiety and depression, and we explored whether serotonin inhibition ameliorates abnormal behavioral phenotypes. Tryptophan hydroxylase 2 (TPH2) is the rate-limiting enzyme for serotonin synthesis in the brain. Parachlorophenylalanine (pCPA), an irreversible inhibitor of TPH2, can significantly reduce serotonin levels in zebrafish after 3 days of continuous treatment (Evsiukova et al, 2021). The results showed that pCPA treatment significantly rescued the overactive locomotor behavior of *nomo1*⁻/⁻ (Fig. EV4B), yet *nomo1*⁻/⁻ zebrafish even exhibited increased interindividual distance than the un-treatment in shoaling test (Fig. EV4A). In contrast, pCPA treatment had no significant effect on the performance of *nomo1*⁻/⁻ in kin preference (Fig. EV4C) and repetitive stereotyped behavior analyses (Fig. EV4D). This suggests that the anxiety-like behavior in *nomo1*⁻/⁻ is also associated with elevated serotonin levels.

## Discussion

A series of genetic etiological analyses have revealed a close association between NOMO1 variants and neuropsychiatric disorders, including autism, epilepsy, and ADHD. Each of these disorders is characterized by distinct behavioral traits that contribute to their diagnostic criteria. In our study, we observed hyperactive locomotor activity in *nomo1*⁻/⁻ zebrafish from 15dpf onwards (Fig. 3), which is a behavioral characteristic associated with anxiety and reminiscent of phenotypes seen in ADHD or epilepsy. In addition, starting from a young age, *nomo1*⁻/⁻ displayed typical autism-like behavioral phenotypes, such as social impairment and repetitive stereotypic behaviors (Fig. 4). Similar phenotypes have been observed in other mutant zebrafish models, such as *shank3b* (Liu et al, 2018), *dyrk1a* (Kim et al, 2017), and *fmr1* (Kim et al, 2014). To further contextualize our findings, we compared them with studies utilizing zebrafish to investigate the role of genes closely related to ASD (Table EV5). Our experimental results, consistent with clinical observations, provide evidence that NOMO1 is a potential candidate gene implicated in neuropsychiatric disorders.

Nodal proteins play a crucial role as inducers of mesendoderm formation, involved in key events such as anterior-posterior axis positioning and left-right axis specialization. Also, their signaling must be blocked in the future neuroectoderm for proper nervous system development (Haffner et al, 2004). Nomo1, along with the nodal signaling antagonists nicalin and TMEM147, forms a complex that effectively inhibits Nodal signaling (Haffner et al, 2004, 2007; Dettmer et al, 2010). Deletion of *nomo1* disrupts the

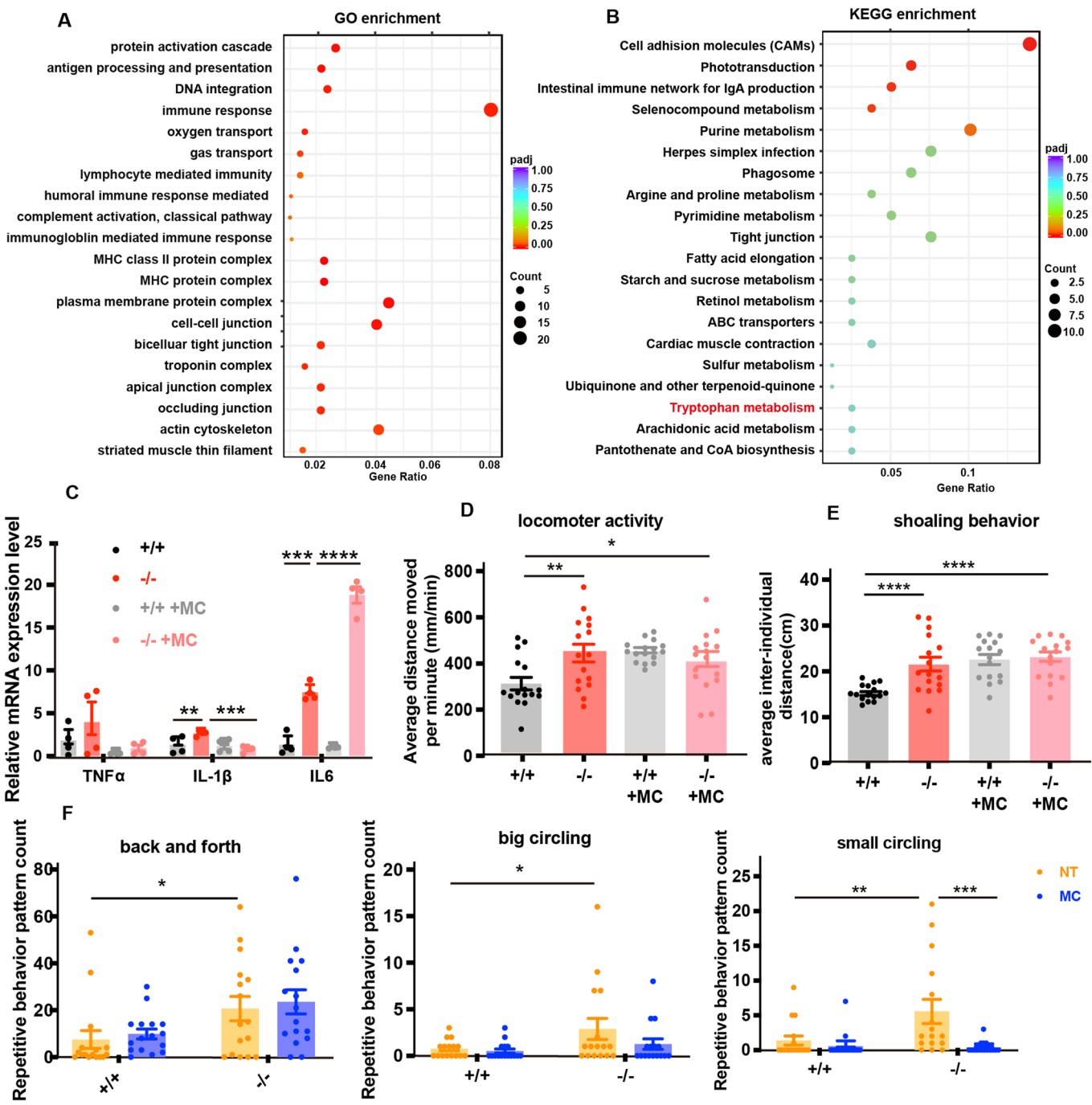

**Figure 6.** *nomo1* **mutant induce brain inflammatory in the brain, MC treatment rescued the small circling behavior.**

(A) GO enrichment analysis of DEGs and (B) KEGG pathway analysis of DEGs. Advanced bubble chart showing the enrichment of DEGs in signaling pathways. The vertical axis indicates the pathway, and the horizontal axis indicates the gene ratio (gene ratio is the ratio of the number of DEGs to the total number of DEGs annotated to the KEGG pathway). The size and color of the bubble represent the number of DEGs enriched in a pathway and the significance of enrichment, respectively. (C–F) (C) Relative expression level of inflammatory cytokines in WT, mutant and MT treatment brains (technical replicates, $N = 4$). Locomotor activity (D) biological replicates, $N = 24$), shoaling behavior (E) (biological replicates, $N = 24$) and three kinds of repetitive behaviors (F) (biological replicates, $N = 24$)) of WT, mutant and MC treatment zebrafish. NT was short for no treatment. Data are analyzed using unpaired t test and presented as the mean ± SEM, $^*P < 0.05$, $^{**}P < 0.01$, $^{***}P < 0.001$ and $^{****}P < 0.0001$. Source data are available online for this figure.

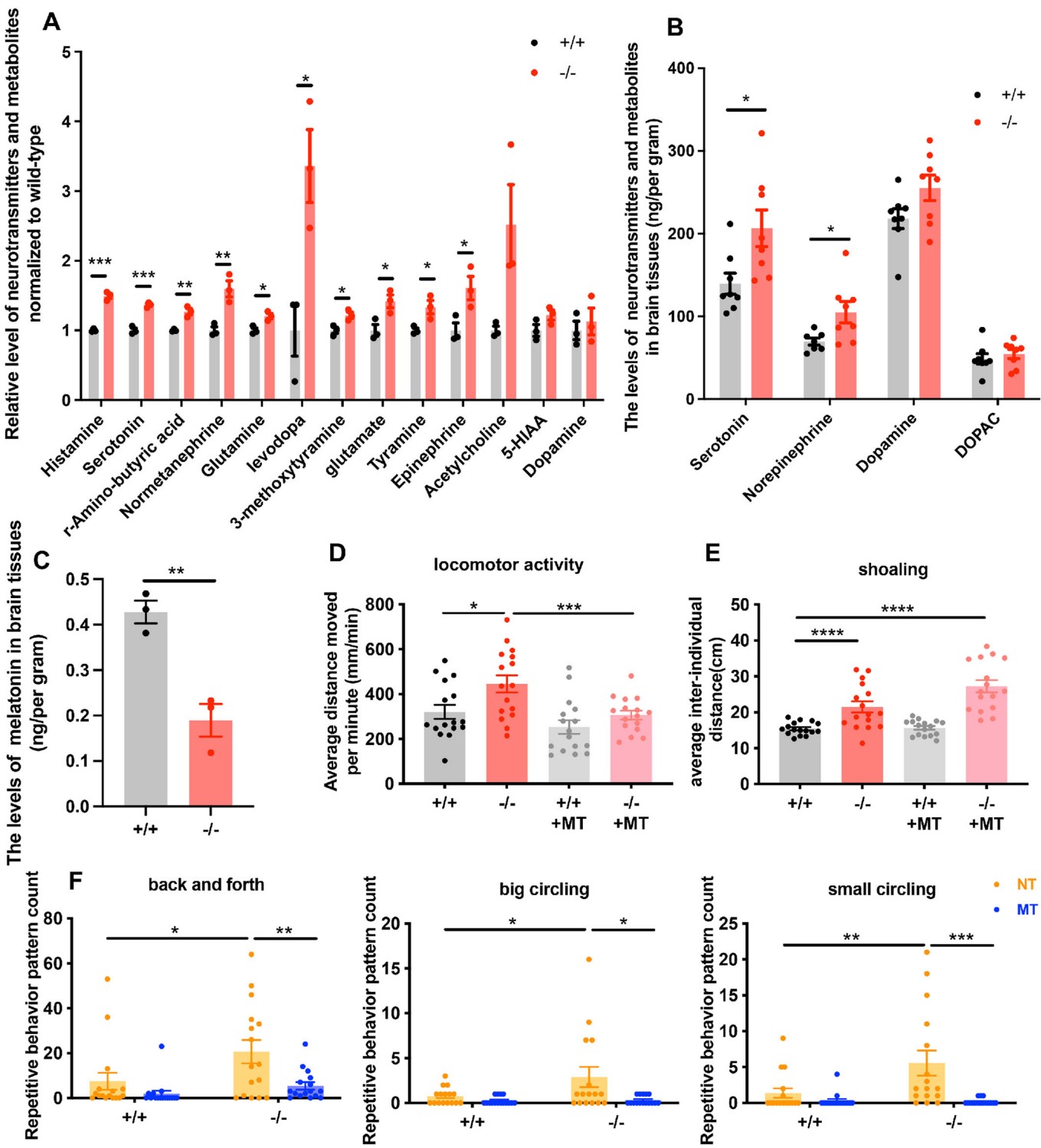

Figure 7. **Melatonin treatment rescues overactive locomotor behavior and repetitive stereotypic behavior in *nomo1$^{-/-}$*.**

(A) Statistical analysis of the SRM/MRM data. The vertical axis denotes the normalized levels of neurotransmitters and metabolites (biological replicates, $N = 3$). (B) Statistical analysis of the HPLC data. The vertical axis denotes the levels of 4 neurotransmitters and metabolites in zebrafish brain (biological replicates, $N = 8$). (C) Melatonin levels of *nomo1$^{-/-}$* and *nomo1$^{+/+}$* (biological replicates, $N = 3$). (D–F) Locomotor activity (D) (biological replicates, $N = 24$), shoaling behavior (E) (biological replicates, $N = 24$), and three kinds of repetitive behaviors (F) (biological replicates, $N = 24$) of WT, mutant and MT treatment zebrafish. NT was short for no treatment. Data are analyzed using unpaired *t* test and presented as the means ± SEM, *$P < 0.05$, **$P < 0.01$, ***$P < 0.001$ and ****$P < 0.0001$. Source data are available online for this figure.

blocking effect of Nodal signaling in the ectoderm, resulting in widespread abnormalities in nervous system development. Transcriptomic analysis revealed that only the expression of *nomo1* was affected within the Nodal signaling pathway, ensuring the normal functioning of Nodal signaling in the mesendoderm. As a result, *nomo1*-deficient zebrafish did not exhibit long-term morphological abnormalities (Fig. EV2C–E), and we didn't observe the chondrodysplasia as in Cao et al (Cao et al, 2018) (Fig. EV2H). In this study, we identified that Nomo1 transcription undergoes early termination at exon 7 (Fig. 1B). In contrast, Cao et al reported an early termination at exon 20. Despite the truncation, the remaining approximately two-thirds of the amino acid chain, including the preserved EMC7-beta-sandw domain, suggests that the abnormally folded protein may still retain some functionality, this abnormal protein variant could potentially exert a different developmental impact compared to the normal Nomo1 protein. In this study, we do observe the developmental processes of mutant embryos are more vulnerable to external environmental factors. Consequently, during the early stages of development, $nomo1^{-/-}$ exhibit transient and mild morphological abnormalities that are not directly attributed to the mutation itself (Fig. EV2A,B,G).

The loss of Nomo1 results in developmental abnormalities in various neuronal populations, including neuronal precursors and motor neurons, during early embryonic development (Fig. 5). Brain functionality follows a developmental trajectory from simplicity to complexity. Prior to the larval stage, the maturation of fundamental motor abilities remains incomplete, and the absence of Nomo1 disrupts the proper development of motor skills in zebrafish. As a result, at 7 dpf, $nomo1^{-/-}$ exhibit significantly reduced locomotor activity (Fig. 2C). As development progresses, basic motor abilities become more refined, and higher-order neural activities and motor control functions begin to emerge. However, the progressive loss of Nomo1 adversely affects an increasing number of neuronal populations (Figs. 1E and 5G,H), leading to a wide array of abnormal behaviors, including notably increased locomotor activity, social impairments, and heightened repetitive and stereotypical behaviors (Fig. 4).

In the adult stage, the absence of Nomo1 results in a structurally fragile brain with significantly reduced mass compared to the wild-type counterparts (Fig. 1D–F). This phenomenon may be attributed to the altered brain stress responses and increased apoptosis associated with the developmental abnormalities caused by Nomo1 deficiency (Fig. 5G,H). In addition, these aberrations contribute to enhanced neuroinflammation (Fig. 6A,B) and dysregulation of various neurotransmitters (Fig. 7A–C).

Melatonin may rescue abnormal behavior through two potential mechanisms. First, it acts by compensating for reduced melatonin synthesis, which is attributed to a decrease in the mesencephalic Ha nucleus, which is an essential part of pineal nerve cells (Fig. 1E). By supplementing melatonin, circadian rhythms can be partially restored, thereby contributing to the amelioration of abnormal behavior. Second, Melatonin is considered a highly neuroprotective substance that exerts cellular protective effects through immune modulation, oxidative stress regulation and other mechanisms (Won et al, 2021). Melatonin treatment inhibited the highly expressed cytokines in $nomo1^{-/-}$ (Appendix Fig. S8), thereby they impeding the stress response of involved cells and mitigating the propensity of the nervous system to sustain severe damage. By aligning with the neuroplasticity of the nervous system, melatonin

supplementation facilitates the restoration of abnormal behavioral phenotypes.

There has been extensive research on the inhibitory effects of minocycline on neuroinflammation (Wang et al, 2020; Henry et al, 2008). In our previous study, we discovered that minocycline can rescue social deficits by suppressing brain inflammation associated with *nde1* deficiency (Zhang et al, 2022). The pathological mechanisms of Nde1 and Nomo1 share certain similarities, as both mutations non-specifically affect early brain development and subsequently induce widespread stress responses and apoptosis in brain tissue. In the case of Nomo1 deficiency, treatment with minocycline restored the repetitive behavior of small circling (Fig. 6F).

According to Bodish et al's classification of repetitive stereotypic behaviors, where stereotypic behaviors are simpler and refer to apparently purposeless, unconscious movements such as, hand clapping, body shaking, arm waving, hand and finger movements, etc. (Lam and Aman, 2007). We believe that small-circle swimming may fall into this category. Another type of repetitive behavior, compulsive behavior, is more complex and refers to behaviors that are repeated or performed according to certain rules, e.g., arranging certain objects in a specific order, constantly checking doors, windows, drawers, counting, etc. We speculate that big circle swimming and back and forth may belong to this category. MC may restore simple stereotyped behaviors to normal levels by suppressing certain specific types of neuroinflammatory responses. Both MC and melatonin have shown improvements in symptoms associated with neuropsychiatric disorders by inhibiting neuroinflammation, indicating that this mechanism may be widely implicated in these disorders. The differential intervention effects of MC in Nomo1 and Nde1 reflect the heterogeneity of the etiology and clinical manifestations of neuropsychiatric disorders.

Increased serotonin levels have been reported to be associated with anxiety-like behaviors, and the application of serotonin inhibitor pCPA has indeed ameliorated the excessive locomotor activity observed in $nomo1^{-/-}$ (Pourhamzeh et al, 2022; Evsiukova et al, 2021) (Fig. EV4). However, pCPA also exacerbated the social deficits in $nomo1^{-/-}$ (Fig EV4). In addition, serotonin levels in the brains of autistic children before the age of five are only one-third of those in typically developing children. In fact, the relationship between serotonin and autism is not fully understood, as both excessively high and low serotonin levels may contribute to abnormal behavioral phenotypes (Pourhamzeh et al, 2022). Moreover, other aberrantly elevated neurotransmitters in *nomo1*-deficient individuals, such as GABA, are also associated with autistic behaviors (Maier et al, 2022). Therefore, the heightened locomotor activity, indicative of anxiety-like behavior, may indeed be related to increased serotonin levels. However, solely suppressing serotonin levels is not a panacea for treating autism.

In conclusion, the deficiency of *nomo1*, a regulatory protein involved in the widely expressed Nodal signaling pathway, leads to diverse developmental abnormalities in neurons during early stages. These abnormalities progressively worsen with ongoing development, triggering a cascade of stress responses and neuroinflammation within the surrounding environment. Consequently, various neurons and neurotransmitters in the brain exhibit aberrant characteristics. Notably, ~70% of individuals with autism display neuroinflammatory responses, indicating that the

amplification of endogenous damage by neuroinflammation and stress may be a prevalent phenomenon in autism.

Promisingly, a prospective open-label trial involving autism children demonstrated that luteolin, a compound known to alleviate brain inflammation in autistic children, improved social activities in some individuals (Tsilioni et al, 2015; Theoharides et al, 2012; Taliou et al, 2013). In addition, a placebo-controlled randomized double-blind study revealed that melatonin significantly ameliorated insomnia symptoms, hyperactivity, and inattention in autism children, although it did not effectively address social deficits (Schroder et al, 2019).

Therefore, further research is warranted to elucidate the mechanisms underlying the extensive brain damage resulting from nomo1 deficiency. Exploring intervention strategies that target stress and inflammation inhibition will not only deepen our understanding of autism but also hold crucial implications for advancing its treatment.

# Methods

## Zebrafish breeding and the generation of *nomo1* mutant zebrafish

Wild-type (WT) zebrafish of the Tuebingen (TU) strain were provided by the zebrafish facility of the Translational Medical Center for Development and Disease, Shanghai Key Laboratory of Birth Defect, Institute of Pediatrics, Children's Hospital of Fudan University. The zebrafish were raised in a circulating water system with a water temperature of 28.5 °C and 14 h of light and 10 h of darkness per day (8:00–22:00, light). Zebrafish breeding, feeding and spawning were conducted strictly in accordance with the Zebrafish Book (http://zfin.org/zf_info/zfbook/zfbk.html).

All procedures comply with the guidelines established by the institutional animal care committee of Children's Hospital of Fudan University, China. All procedures were approved by the Institutional Animal Care Committee of Children's Hospital of Fudan University, China (approval number: 2020-201).

The CRISPR/Cas9-mediated editing method was performed using standard procedures (Hwang et al, 2013; Mali et al, 2013). A synthetic specific guide RNA (sgRNA) (sequence: 5'-GGG CTA TGA TGT CTC TGG AG- 3') and Cas9 mRNA (concentrations of 30 ng/μL and 300 ng/μL, respectively) in a total volume of 3 nL were coinjected into every WT zebrafish embryo at the single-cell stage. Genomic DNA was extracted, and genotyping samples were screened for the mutation frequency by comparison with WT zebrafish samples. The primer sequences used for genotyping are shown in Table EV1. The mutant chimeric zebrafish were mated with the TU strain to purify the background and obtain $nomo1^{+/-}$ zebrafish. Male and female $nomo1^{+/-}$ zebrafish were crossed to acquire $nomo1^{+/+}$, $nomo1^{+/-}$, and $nomo1^{-/-}$ littermates. We collected multiple batches of littermates produced by $nomo1^{+/-}$ zebrafish for the phenotypic analysis to obtain a sufficient number of embryos.

## Quantitative real-time RT-qPCR

Total RNAs were extracted from embryos, heads and the brain tissues of zebrafish at different developmental stages using TRIzol reagent (Ambion, USA). Genomic DNA was removed by DNase I, and total RNA (1 μg) was reverse transcribed using a PrimeScript cDNA Synthesis Kit (TaKaRa, Japan). RT-qPCR was conducted with a LightCycler® 480 apparatus (Roche, Germany) and Super-Real PreMix Plus (Tiangen, China) according to the manufacturer's instructions. Each biological $n$ has four technical replicates, β-actin gene was used as a housekeeping gene. The fold changes in RNA levels were calculated using the ΔΔCt method. The RT-qPCR primer sequences are listed in Table EV1.

## WISH

The targeted DNA was cloned into the pGEM-T Easy vector, and probes were synthesized using a linearized plasmid through in vitro transcription with the DIG-RNA labeling Kit (Roche, Austria). The related primers of synthetic probes are shown in Table EV1. Embryos of WT and mutant zebrafish were collected at 48 h post fertilization (hpf) and fixed with 4% paraformaldehyde at 4 °C overnight. WISH was performed as previously described (Thisse and Thisse, 2008), and images were captured and processed using a Leica 6000 microscope.

## Behavioral assays in mutant zebrafish

### Locomotion and thigmotaxis tests
Behavioral assays of larval zebrafish were performed at 28.5 °C in 24-well plates (Fig. 2G), and the inner diameter of each well was 18 mm, providing the larvae sufficient space to swim. The 24-well plates were then placed in a Zebrabox (ViewPoint Life Sciences, Lissieu, Calvados, Lower Normandy Region, France) that recorded videos tracking the larval zebrafish. The experimental procedure consisted of 55 min of continuous illumination with light at an intensity of 100 lx and two 10-min light–dark transition cycles for a total time of 75 min to elicit a photomotor response (PMR) (Fig. 2A). The experiment examined both spontaneous movement and changes due to lighting transitions. The data were quantified with ZebraLab software (ViewPoint Behavior Technology, France), the video rate was set to 25 frames per second (fps), and the frames were pooled into 1-min time bins. The threshold was set to 29, a suitable level to accurately detect the trajectory of larval zebrafish in motion.

Zebrafish at 15 days post fertilization (dpf), 30 dpf and 2 months post fertilization (mpf) swam freely in the open field at 28.5 °C. Behavioral recording began after an adaptation period (1–2 min) when the zebrafish acclimated to the environment. Zebrafish at 15 dpf were examined in a 35-mm diameter dish since they were a smaller size (Fig. 3A). Zebrafish at 30 dpf and 2 mpf were examined in a novel tank (inner dimensions, 30 × 30 × 20 cm) (Fig. 3C). The collected data were exported using ZebraLab software.

### Social and repetitive behavior tests
The individual social behavior (social preference behavior) and group social behavior (shoaling behavior) of 2-mpf zebrafish were assessed at 28.5 °C. A single zebrafish was placed on one side of a standard mating tank (inner dimensions, 21 × 10 × 7.5 cm), and another six WT zebrafish were placed on the other side of the mating tank and separated from the single zebrafish by a transparent plastic plate to examine social preference behavior (Fig. 4A). Region 1 was regarded as a social area, whereas Region 2 was regarded as a nonsocial area, and the experiment lasted for 30 min. Behavioral recording began

after an adaptation period (1–2 min) when the zebrafish acclimated to the tank. The behavior of the zebrafish was quantified as a distribution of distances or regions adjacent to the group. The ratio of the time the zebrafish stayed in the social area and the distance spent away from the social area directly reflects the social activity of a single juvenile zebrafish.

For the shoaling test, six WT zebrafish (or six $nomo1^{-/-}$ zebrafish) were acclimated to a novel tank (inner dimensions, $30 \times 30 \times 20$ cm) (Fig. 4F). A camera recorded the trajectory of the experimental zebrafish over 30 min, and the adaptation period was 1–2 min. The indicator of interindividual distance was used to assess the average distance between each zebrafish in the shoal.

We observed different types of repetitive behavior (back-and-forth motions, stereotypic movement and large circular movement) in the $nomo1^{-/-}$ zebrafish (Fig. 4H) when we examined the spontaneous movement of juvenile zebrafish. All the repetitive behavior tests began after an adaptation period (1–2 min) when the zebrafish acclimated to the tank. Back-and-forth motion was defined as moving one time on one edge or adjacent edges of the tank and returning to the origin. Stereotypic movement referred to the repeated movement of the zebrafish in a small area, where the maximum movement distance from the beginning to the end was less than 30 mm and continuous swimming time was greater than 5 s. Large circular movement referred to the swimming of the zebrafish in a counterclockwise or clockwise circle along the edges of the tank. After the stereotypic movements were defined, the data were obtained objectively by a computer program within the experimental period.

## Preparation of paraffinized sections and HE staining

The brain tissues of adult zebrafish (3 mpf) were completely removed under a microscope and immersed in 4% paraformaldehyde for 24 h. Tissues were dehydrated and transparentized with the following conditions and protocol: 70% ethanol, 30 °C, 30 min; 95% ethanol, 30 °C, 10 min; 95% ethanol, 30 °C, 10 min; 100% ethanol (I), 30 °C, 10 min; 100% ethanol (II), 30 °C, 10 min; xylene (I), 30 °C, 30 min; and xylene (II), 30 °C, 30 min. The tissues were waxed and embedded using a paraffin embedding station (Leica EG1150H) for 3 h at 65 °C. Then, a microtome (Leica RM2235) was used to produce continuous slices at a thickness of 4 µm. The slices were floated in 40 °C warm water to flatten their surfaces, and then baked in a 60 °C oven. HE staining was performed by staining with hematoxylin and eosin for 5 min each. Photographs were taken with a Leica 205 C microscope.

To quantitatively analyze cells within the Vp, PM, and PPp nuclei in the forebrain, the following method was employed: a circular region of interest (ROI) was positioned over the corresponding area, and after converting the image to 8-bit with image J software, cells were outlined by adjusting the threshold. Measure tool was then used to obtain the percentage of cell signal within the ROI. For quantitative analysis of Ha nuclei of telencephalon with pineal gland, and Vam, VamG nuclei of hindbrain, the nuclei area was outlined using a ticking tool in image J, Measure tool was used to obtain the area data in pixels.

## Alizarin Red and Alcian Blue staining

For a fixed sample, all specimens (8 dpf larvae) were anesthetized using tricaine and then fixed in 4% paraformaldehyde for 1 hour at room temperature. They were subsequently washed and dehydrated with 50% ethanol for 10 minutes at room temperature. To the larvae, 0.5% Alizarin Red (Yeasen, 60504ES25, Shanghai, China) or 0.1% Alcian Blue staining (Servicebio, G1027, Wuhan, China) Solution was added, and staining was done overnight at room temperature. After elution of the staining solution by gradient ethanol, a final concentration of 1.5% $H_2O_2$ and 1% KOH bleach solution was added for bleaching. Final decolorization was performed overnight with 20% glycerol and 0.25% KOH solution, followed by 50% glycerol and 0.25% KOH solution, respectively. The larvae were observed under a Leica dissecting microscope.

## Generation of transgenic line

Transgenic zebrafish line Tg (*HuC: RFP*) is from Xu Wang, Tg (*Gad1b: EGFP*) and Tg (*Vglut2a: dsRed*) were kindly provided by Professor Jie He. Transgenic zebrafish Tg (*nomo1^{-/-}, Huc:RFP*), Tg (*nomo1^{-/-}, Gad1b: EGFP*) and Tg (*nomo1^{-/-}, Vglut2a: dsRed*) were generated by two generation cross and genotyping.

## Fluorescence signal observation and quantification

Fluorescence signal observation and Imaging were performed using a confocal microscope (TCS-sp8, Leica) with a ×20 objective. Split the fluorescent signal using Image-Color-Split Channels tool of image J software. Adjust the threshold, select the signal, and use the measure tool to get the quantitative Integrated Density (IntDen) for comparative analysis.

## Transcriptomics

The total RNA was extracted from each sample of brain tissue from 2-mpf zebrafish using TRIzol reagent (Ambion, USA) according to the manufacturer's instructions. RNA degradation and contamination were monitored on 1% agarose gels. The RNA concentration was measured using a Qubit® RNA Assay Kit with a Qubit®2.0 fluorometer (Life Technologies, CA, USA), and RNA integrity was assessed using an RNA Nano 6000 Assay Kit of the Bioanalyzer 2100 system (Agilent Technologies, CA, USA). Three micrograms of RNA per sample was used as the input for RNA sample preparation. Sequencing libraries were generated using a NEBNext® Ultra™ RNA Library Prep Kit for Illumina® (New England Biolabs, USA) according to the manufacturer's recommendations, and index codes were added to attribute sequences to each sample. PCR products were then purified (AMPure XP system), and library quality was assessed using the Agilent Bioanalyzer 2100 system. Sequencing fragment data detected using a high-throughput sequencer was converted from image data into sequence data (reads) containing sequence information from each sequenced fragment and its corresponding sequencing quality by CASAVA base recognition. High-quality reads were aligned to the zebrafish reference genome (GRCz11) using HISAT2 v2.0.5. Differentially expressed genes (DEGs) were identified using the DESeq2 R package (1.16.1), which uses statistical methods to determine differential expression from digital gene expression data using a model based on a negative binomial distribution. The resulting $P$ values were adjusted using the Benjamini and Hochberg approach for controlling the false discovery rate. Genes with an adjusted $P$ value < 0.05 according to DESeq2 were defined as DEGs. In

addition, a Gene Ontology (GO) enrichment analysis of the DEGs was implemented by the clusterProfiler R package, which corrected the gene length bias. GO terms with corrected *P* values < 0.05 were considered significantly enriched in DEGs. The Kyoto Encyclopedia of Genes and Genomes (KEGG) database is a resource used to understand high-level functions and utilities of biological systems, such as a cell, an organism and an ecosystem, and it is specifically focused on analyzing large-scale molecular datasets generated by genome sequencing and other high-throughput experimental technologies (http://www.genome.jp/kegg/). We used the cluster-Profiler R package to assess the statistically significant enrichment of DEGs in KEGG pathways.

## Targeted metabolomics analysis of neurotransmitter

Juvenile zebrafish were frozen in liquid nitrogen and placed on ice. Under the microscope, the brains were directly removed with a surgical blade, and ten brains were placed in an Eppendorf (EP) tube. Selective/multiple reaction monitoring assays (SRM/MRM), which is based on liquid chromatography-tandem mass spectrometry (LC-MS/MS), has been used to simultaneously detect neurotransmitters in animals. LC-MS/MS can determine absolute quantitative amounts of target metabolites with strong specificity and high sensitivity and accuracy (Huang et al, 2014).

Brain tissue samples were added to 1 mL of methanol/acetonitrile/ultrapure water (2:2:2, v/v/v), vortexed and ultrasonicated. After incubation at −20 °C for 30 min, the precipitated proteins were centrifuged at 14,000 rcf for 4 min at 4 °C. The supernatant was removed and dried in vacuo. For spectrometric detection, 100 µL of an acetonitrile/water solution (1:1, v/v) were used to reconstitute the pellet, which was centrifuged at 14,000 rcf for 4 min at 4 °C, after which the supernatant was collected for analysis. Samples were separated using an Agilent 1290 Infinity LC Ultra Performance LC System. Mass spectrometry was conducted in negative ion mode using a 5500 QTRAP mass spectrometer (AB SCIEX) for analysis. Data on stability and repeatability were evaluated using cluster and statistical analyses.

## HPLC

High-performance liquid chromatography (HPLC) was also employed to examine the levels of norepinephrine, dopamine (DA), 3,4-dihydroxyphenylacetic acid (DOPAC), and serotonin (5-HT) in the brains of juvenile zebrafish. The samples were suspended in ice-cold PBS (20 µL/sample) and ground completely. Then, the samples were centrifuged at 11,900 rpm at 4 °C for 10 min, and the protein content of 1 µL of the supernatant from each sample was quantified. The remaining supernatants were added to 2 µL of perchloric acid (0.2 N) and centrifuged again at 11,900 rpm at 4 °C for 10 min. The supernatant was collected and stored at −80 °C. The HPLC analysis was performed using an Agilent 1200 HPLC system (Agilent, USA) with Antec DECADE SDC electrochemical detector (Antec, the Netherlands). The expression levels of neurotransmitters were normalized to the protein content.

## Drugs treatment

Minocycline (Beyotime, ST1528, Beijing, China) was dissolved in ultrapure water to prepare a stock solution. 2-mpf WT and *nomo1*

deficiency zebrafish were treated with 10 mM minocycline for 4 weeks. Fresh minocycline water was replaced every day. Under the normal maintenance condition, the surface light intensity of the fish tank was turned down by 100 lux for preventing minocycline's full decomposition by light.

Melatonin (Energy Chemical, E120134, Shanghai, China) was dissolved in DMSO to prepare a 2 M stock solution. 2-mpf WT and *nomo1* deficiency zebrafish were treated with 5 µM MT every night for 4 weeks.

pCPA (MedChemExpress, HY-B1368, Monmouth Junction, NJ, USA) was dissolved in DMSO to prepare a 20 mg/L stock solution. 2-mpf WT and *nomo1* deficiency zebrafish were treated with 2 mg/L pCPA for 72 h. The MC, MT and pCPA working solution was diluted to the appropriate concentration with system water prior to the experiments.

## Experimental design and statistical analysis

The experiment was conducted in a blinded manner. The sample preparer numbered the samples, and the experimental analysis operator was unaware of the sample type. The experimental data in this study were analyzed and mapped with GraphPad Prism 9.0 software. Values are presented as the means ± SEM. Differences between two groups were analyzed using unpaired Student's *t* test, and *P* values < 0.05 indicated a significant difference.

# Data availability

Sequencing data reported in this paper has been deposited to SRA (PRJNA806676) and is publicly available.

# Peer review information

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

## Acknowledgements

We thank all members of the Qiang Li laboratory for their support, and we thank the Translational Medical Center for Development and Disease of Children's Hospital of Fudan University in China. We thank Dr. Ning Guo for the advice on behavioral analyses and Chenwen Zhu for the technical discussions related to the experiments. This study was supported by grants from the National Natural Science Foundation of China (NSFC, no. 81771632 and no. 81271509) to Qiang Li, the Natural Science Foundation of Shanghai (grant No. 21ZR1410100) to Qiang Li, and the National Natural Science Foundation of China (NSFC, no. 82201310) to Qi Zhang.

## Author contributions

**Qi Zhang**: Conceptualization; Funding acquisition; Validation; Investigation; Visualization; Methodology; Writing—original draft; Project administration; Writing—review and editing. **Fei Li**: Data curation; Software; Formal analysis; Investigation; Methodology; Writing—original draft. **Tingting Li**: Resources; Formal analysis; Validation. **Jia Lin**: Resources; Formal analysis. **Jing Jian**: Investigation; Visualization. **Yinglan Zhang**: Formal analysis; Investigation. **Xudong Chen**: Software; Investigation. **Ting Liu**: Resources; Data curation; Investigation. **Shenglan Gou**: Investigation; Visualization. **Yawen Zhang**: Data curation; Formal analysis. **Xiuyun Liu**: Resources; Investigation. **Yongxia Ji**: Resources; Methodology. **Xu Wang**: Resources; Methodology. **Qiang LI**: Conceptualization; Supervision; Funding acquisition; Project administration; Writing—review and editing.

## Disclosure and competing interests statement

The authors declare no competing interests.

# Expanded View Figures

**Figure EV1.  Protein sequence of zebrafish WT and mutant Nomo1 protein.**

(**A, B**) Amino acid sequences of WT and mutant Nomo1 protein.

## A WT Nomo1 protein sequence

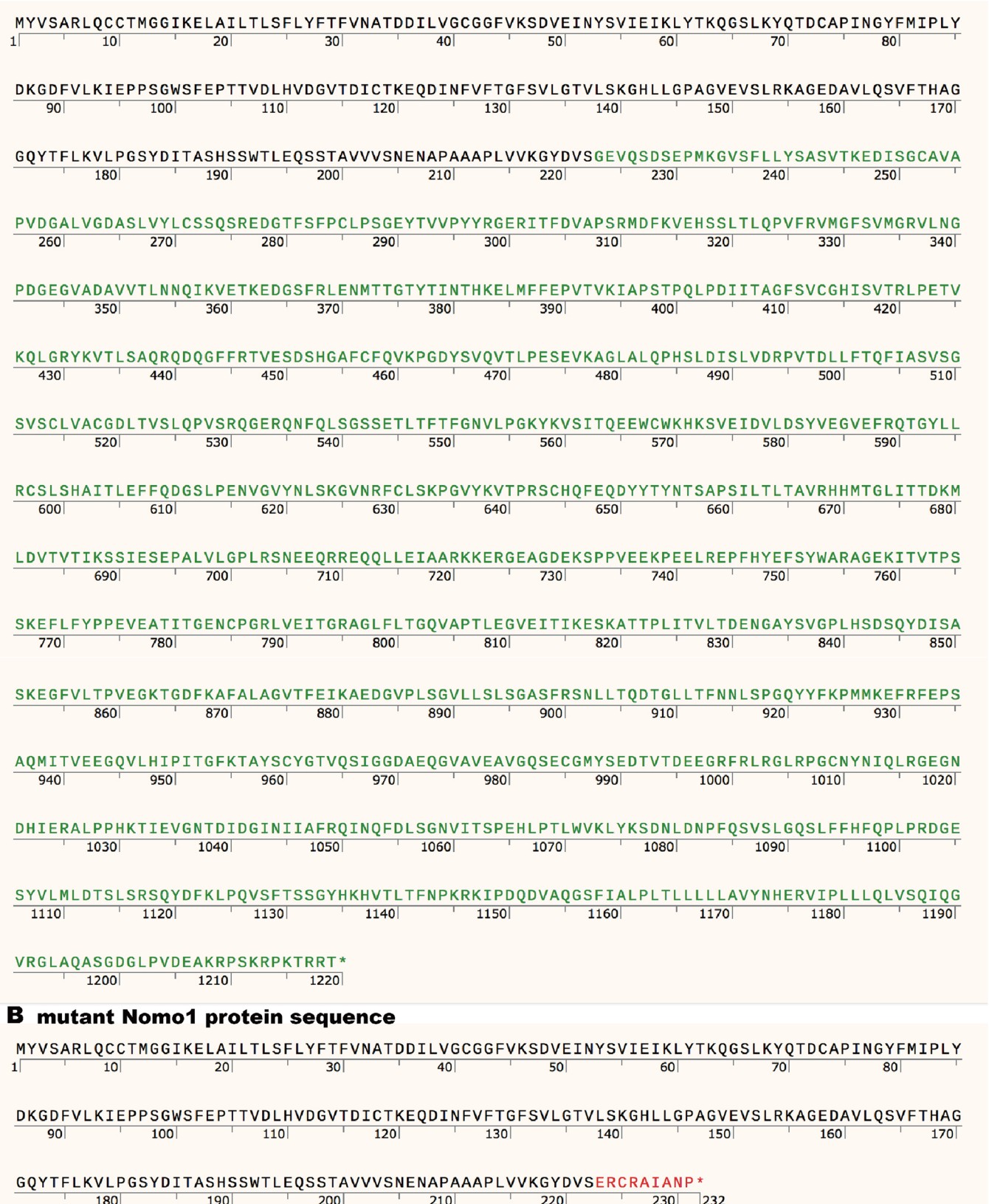

## B mutant Nomo1 protein sequence

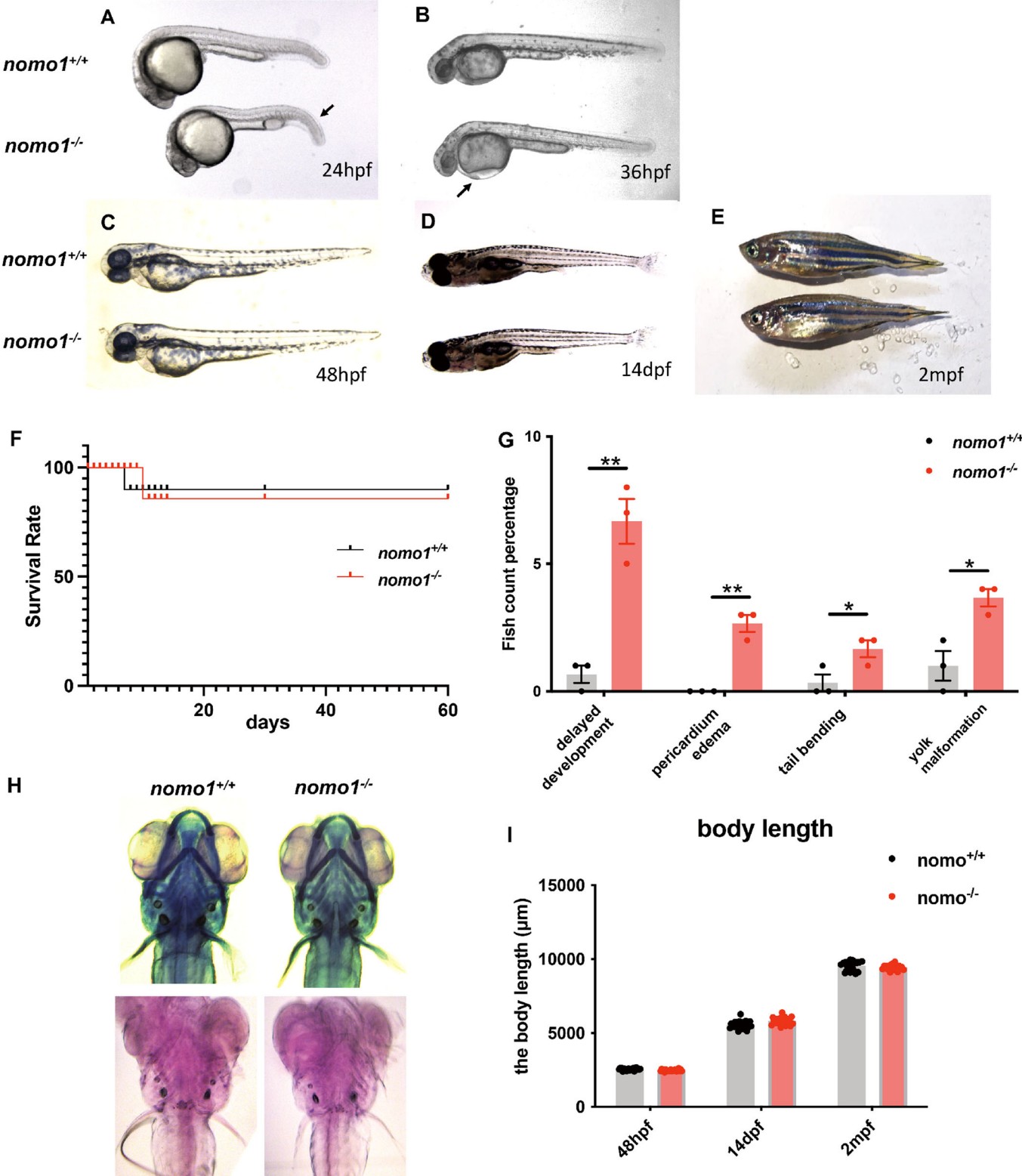

◀   **Figure EV2.   Morphological analysis of WT and *nomo1* mutant zebrafish during developmental stages.**

(**A–E**) The morphology of WT and *nomo1* mutant zebrafish at 24 hpf (**A**), 36 hpf (**B**), 48 hpf (**C**), 14 dpf (**D**) and 2 mpf (**E**). (**F**) survival rate of *nomo1*$^{+/+}$ and *nomo1*$^{-/-}$ from 0 day to 60 days (biological replicates, $N = 48$). (**G**) Abnormal morphology of *nomo1*$^{+/+}$ and *nomo1*$^{-/-}$ at 24 dpf including developmental delay, tail bending, pericardium edema and yolk malformation (biological replicates, $N = 3$). (**H**) Alcian Blue and Alizarin Red staining showed Skeletal and lower jaw development of *nomo1*$^{+/+}$ and *nomo1*$^{-/-}$ at 7dpf. (**I**) The body length of *nomo1*$^{+/+}$ and *nomo1*$^{-/-}$ zebrafish at 48 hpf, 14 dpf and 2 mpf (biological replicates, $N = 36$). Data information: Data are analyzed using unpaired t test and shown as the mean ± SEM. $^{*}P < 0.05$, $^{**}P < 0.01$.

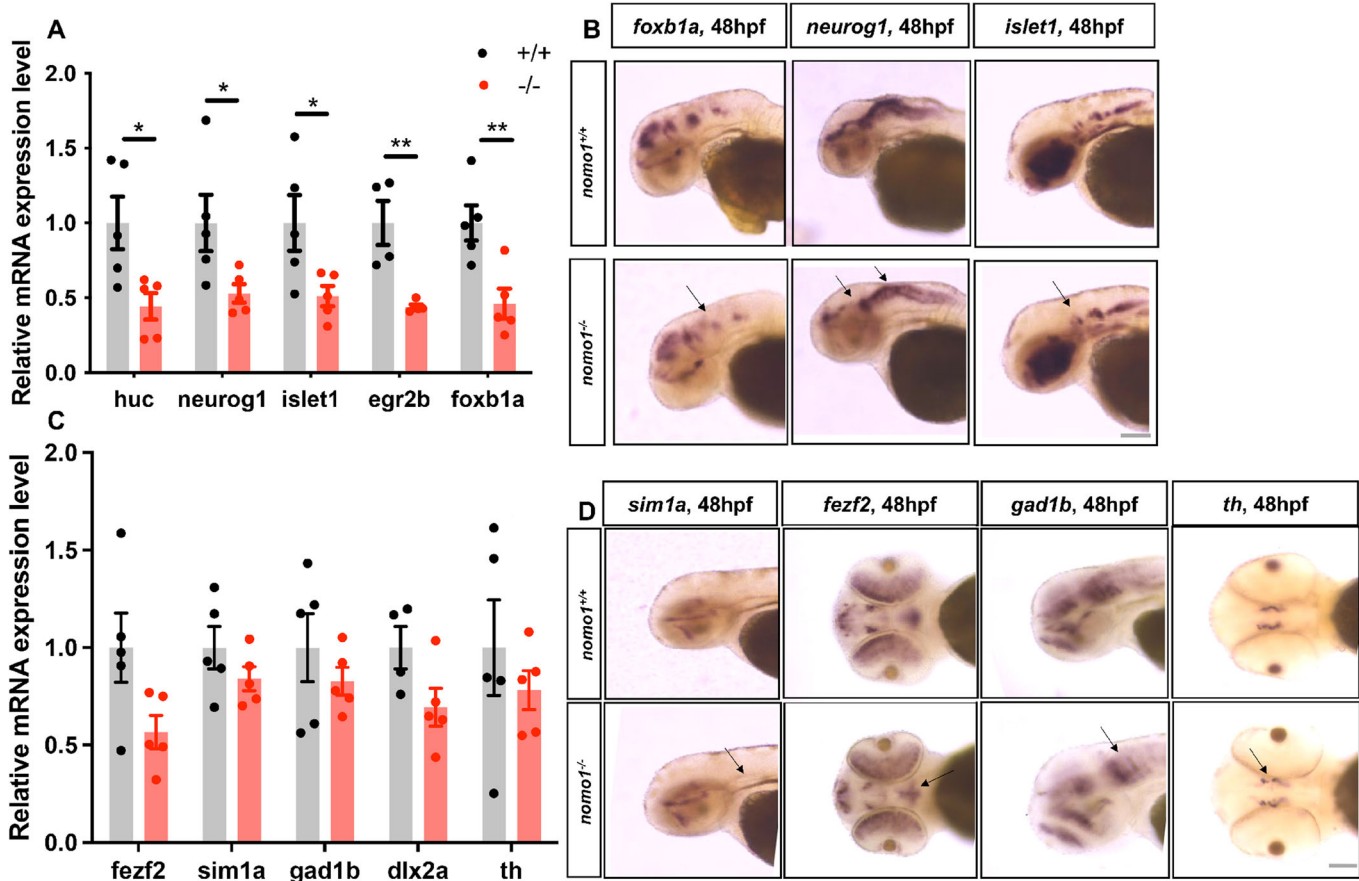

**Figure EV3. Expression analysis of neurodevelopment related genes in WT and Nomo1 mutant zebrafish.**

(A–D) (A, C) The expression level of neurological genes in brain of 48-hpf wt and $nomo1^{-/-}$ (biological replicates, $N = 5$). (B, D) Expression of neurological genes were detected using WISH, arrow heads indicate the expression were inhibited. Data are analyzed using unpaired $t$ test and presented as the means ± SEM. $*P < 0.05$ and $**P < 0.01$. Scale bar $= 100\ \mu m$.

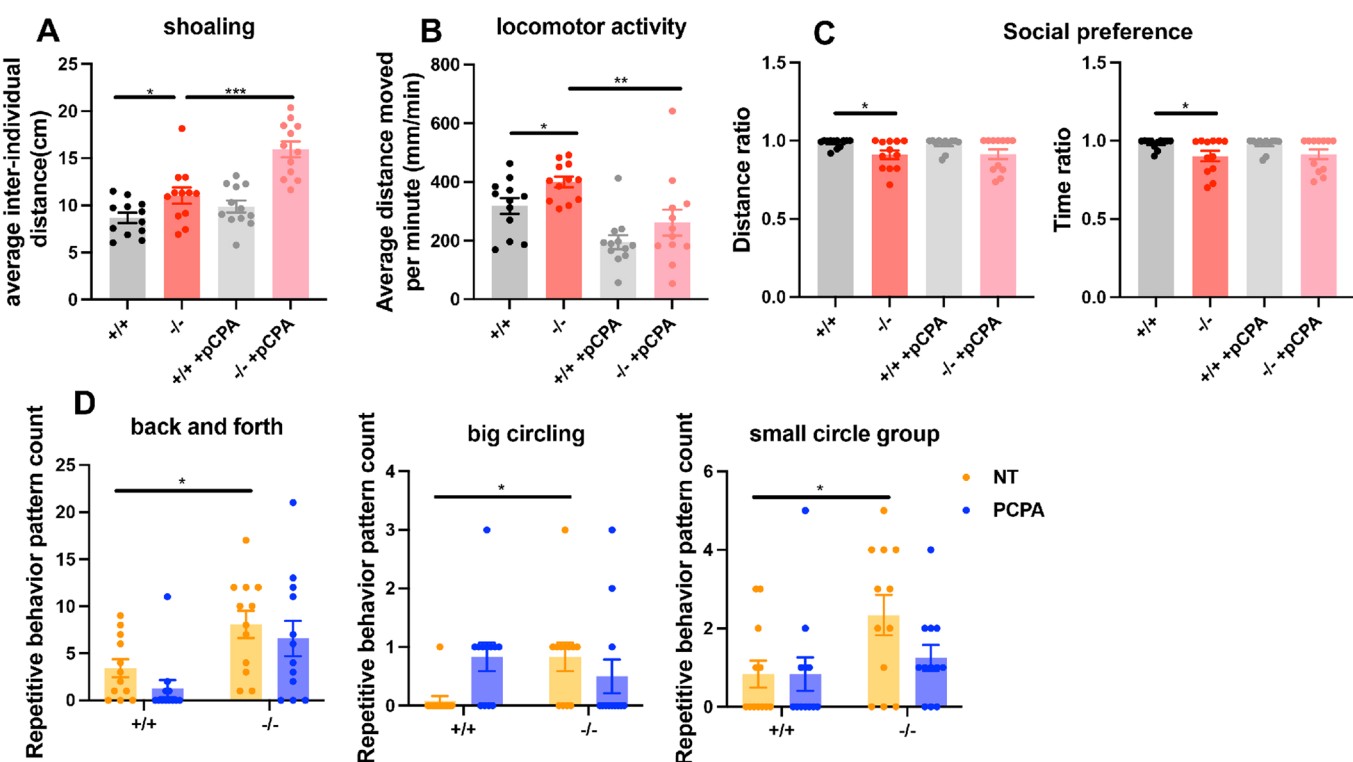

**Figure EV4. pCPA treatment rescued the overactive locomotion and exacerbates social deficits of *nomo1*−/−.**

(A–D) Shoaling behavior (A) (biological replicates, $N = 12$) locomotor activity (B) (biological replicates, $N = 12$), social preference (C) (biological replicates, $N = 12$) and three kinds of repetitive behaviors (D) (biological replicates, $N = 12$) of WT, mutant and pCPA treated zebrafish. NT no treatment. Data are analyzed using unpaired *t* test and presented as the means ± SEM, $^*P < 0.05$, $^{**}P < 0.01$, $^{***}P < 0.001$.

