## [Peer Review File · EMBO Reports]

Nomo1 deficiency causes autism-like behavior in zebrafish

Qi Zhang, Fei Li, Tingting Li, Jia Lin, Jing Jian, Yinglan Zhang, Xudong Chen, Ting Liu, Shenglan Gou, Yawen Zhang, Xiuyun Liu, Yongxia Ji, Xu Wang, and Qiang Li

DOI: [10.15252/embr.202256109](https://doi.org/10.15252/embr.202256109)

Corresponding author(s): [Qiang Li \(liq@fudan.edu.cn\)](mailto:liq@fudan.edu.cn)

Review Timeline:

Submission Date:	8th Sep 22
Editorial Decision:	17th Oct 22
Revision Received:	11th Jul 23
Editorial Decision:	5th Sep 23
Revision Received:	30th Nov 23
Accepted:	6th Dec 23

Editor: *Esther Schnapp / Martina Rembold*

Transaction Report:

Dear Prof. LI,

Thank you for the submission of your manuscript to EMBO reports. We have now received the full set of referee reports that is pasted below.

As you will see, the referees acknowledge that the findings are potentially interesting. However, they also point out that the mechanistic links between the *nomo* mutation, the phenotypes and the rescue by melatonin remain unclear. It would certainly strengthen the study if such links could be provided or possible hypotheses could be strengthened. The referees have several suggestions for how this could be done, and I think these suggestions are good. Please let me know in case you disagree or if you would like to discuss the revisions further, and we can do that, also in a video chat, if you like. All technical comments must be addressed.

I would thus like to invite you to revise your manuscript with the understanding that the referee concerns must be fully addressed and their suggestions taken on board. Please address all referee concerns in a complete point-by-point response. Acceptance of the manuscript will depend on a positive outcome of a second round of review. It is EMBO reports policy to allow a single round of major revision only and acceptance or rejection of the manuscript will therefore depend on the completeness of your responses included in the next, final version of the manuscript.

We realize that it is difficult to revise to a specific deadline. In the interest of protecting the conceptual advance provided by the work, we recommend a revision within 3 months (17th Jan 2023). Please discuss the revision progress ahead of this time with the editor if you require more time to complete the revisions.

- 1) A data availability section providing access to data deposited in public databases is missing. If you have not deposited any data, please add a sentence to the data availability section that explains that.
- 2) Your manuscript contains statistics and error bars based on $n=2$. Please use scatter blots in these cases. No statistics should be calculated if $n=2$.

3) We replaced Supplementary Information with Expanded View (EV) Figures and Tables that are collapsible/expandable online. A maximum of 5 EV Figures can be typeset. EV Figures should be cited as 'Figure EV1, Figure EV2' etc... in the text and their respective legends should be included in the main text after the legends of regular figures.

5) a complete author checklist, which you can download from our author guidelines <https://www.embopress.org/page/journal/14693178/authorguide>. Please insert information in the checklist that is also reflected in the manuscript. The completed author checklist will also be part of the RPF.

6) Please note that all corresponding authors are required to supply an ORCID ID for their name upon submission of a revised manuscript (<https://orcid.org/>). Please find instructions on how to link your ORCID ID to your account in our manuscript tracking system in our Author guidelines

<<https://www.embopress.org/page/journal/14693178/authorguide#authorshippingguidelines>>

8) At EMBO Press we ask authors to provide source data for the main and EV figures. Our source data coordinator will contact you to discuss which figure panels we would need source data for and will also provide you with helpful tips on how to upload and organize the files.

- the name of the statistical test used to generate error bars and P values,
- the number (n) of independent experiments (please specify technical or biological replicates) underlying each data point,
- the nature of the bars and error bars (s.d., s.e.m.),
- If the data are obtained from n {less than or equal to} 2, use scatter blots showing the individual data points.

I look forward to seeing a revised form of your manuscript when it is ready.

Yours sincerely,

Esther Schnapp, PhD
Senior Editor

Referee #1:

In this study, Zhang et al. generated a *nomo1*^{-/-} zebrafish and examined the role of *nomo1* in neurodevelopment, neuroinflammation and behaviour. They found that the *nomo1* mutant zebrafish exhibit behavioural defects include autism-like repetitive behaviors and social defects. At the molecular level, they showed that *nomo1* deficiency affects neuronal development (downregulation of *huc*, *islet1*, *neurog1*), neurotransmitter metabolism and that melatonin treatment significantly rescued the certain behaviours of *nomo1*^{-/-} zebrafish.

Overall, this manuscript provides new insights into the role of *nomo1* in brain development and behaviour. The mechanistic links between *nomo1*, developmental defects, behavioural phenotypes and melatonin rescue remain somewhat weak. Experiments to strengthen these links in both larval and adult stages can significantly strengthen the mechanistic conclusions of the manuscript.

Major comments:

Line 299-300 "these morphological abnormalities of *nomo1*^{-/-} zebrafish gradually became less noticeable during development (Fig. S2C-E)" - Is it because those with abnormalities die prematurely? A survival curve can be helpful. Do homozygous *nomo1*^{-/-} survive to adult stage?

Do *nomo*^{+/-} fish display any abnormalities?

A low percentage of morphological abnormalities were observed in *nomo1*^{-/-} zebrafish according to the authors (Suppl. Figure 2). Representative figures in S2 do not show signs of bone defects - yet it is confusing that in the discussion the authors state at line 496-498 that locomotion defects in 7 dpf may be associated with impaired bone development in 7 dpf *nomo1*^{-/-} larvae. Were skeletal defects observed in *nomo1*^{-/-} zebrafish? Skeletal deformities can impact behaviour and melatonin plays a role in skeletal development- the rescue of behaviour may thus be misinterpreted. Additionally, Cao et al., 2018 (ref 11) showed that zebrafish *nomo* homozygous mutants exhibited chondrodysplasia.

The *nomo1*^{-/-} brain in Fig1D seems to be different in size. Importantly, the olfactory bulb and forebrain are clearly reduced in size in *nomo1*^{-/-} Fig 1D. Please provide the data for the brain size and a better representative brain for *nomo1*^{-/-} is required if there are no significant differences in *nomo1*^{-/-} and control brain size. Please also include section and staining of the forebrain in Fig. 1E.

The authors observe a difference in the mass of the brain tissue in the model. Did the authors observe an increase in apoptosis? Are there specific regions of the brain affected in terms of reduced mass?

Important markers are found down-regulated in 48 hpf *nomo1*^{-/-} fish such as *huc*, *neurog1*, *islet1* (Fig. 5). Are they still down-regulated in adult fish? Are there less neurons in larval brain? Brain sections could have been more thoroughly analysed at larval stages.

Was there a difference in expression of the motor neuron marker *islet-1* in the spinal cord? Alterations in motor neuron number can also impact motor behaviour.

Figure legends do not state the statistical test performed for each analysis, these need to be stated. Particularly, in the methods (line 267), the authors state the groups were analysed using paired Student's t-test and such a statistical test is not appropriate for most datasets presented in this study. An unpaired t-test should be used when comparing WT vs mutant data for instance.

The authors state that "repetitive stereotypic behaviors of small circle swimming were restored to normal levels (Fig. 6H)" with MC treatment, but not the other repetitive behaviors. Any explanation for why there is a selective restoration in behavioural defects with MC treatment?

Is melatonin decreased in *nomo1*^{-/-} fish? Can melatonin be evaluated as well by HPLC to support their claim?

The mechanistic link between *nomo1*, reduced expression of neuronal progenitor and mature neuron markers at 48 hpf, abnormal behavioural phenotypes and increase in inflammatory pathways, serotonin and ASMT/melatonin remain weak (including in the discussion). Can melatonin restore behavioural defects in 7 dpf larvae?

How can defects in social behaviour be explained upon *nomo1* deficiency? Neuronal network in *nomo1* brain at larval and adult stages can be thoroughly examined by immunostaining.

Minor comments:

In the introduction, the authors mention the role of *NOMO1* in the inhibition of the Nodal pathway. What is the significance of this role in the study? Were the genes in the Nodal pathway perturbed in the transcriptomic analyses? This should be commented.

At what age (be more specific than juvenile) was transcriptomic analysis performed?

Only dpf is defined in the method section, please define hpf and mpf as well.

Genes should be italics (line 273)

Please use "did not" instead of "didn't"

7dpf are larvae not embryo (line 278, line 496)

Graphs could be better presented as dot plots, with the average and appropriate error bars indicated

Line 321, "confirming the specificity of the phenotype" - What is meant by this strong statement? What specific phenotype is the authors referring to here?

The Minocycline-treated animals were kept in a light-depleted environment. Could this have an influence on motor behaviour?

Explaining the thigmotaxis test and its choice - could be beneficial in understanding the clinical relevance of the read-out.

Disparities in the colour of the graphs can be a bit confusing (for example: Figure 4D, E, G, I; Figure 7.H)

Figure 3B, statistics should be included.

Referee #2:

Dear authors, co-reviewers and editors,

the manuscript entitled „*NOMO1* deficiency results in abnormal behavior and disturbed serotonin metabolism, melatonin supplementation exhibited rescue effect" by Zhang and Li et al., describes the generation and analysis of a *nomo1* deficient zebrafish line. The motivation to analyze *nomo1* is based on the chromosomal localization of the human gene in 16p13.11, a region which is involved in several neuropsychiatric disorders. The authors demonstrate that loss of *nomo1* results in mild morphological abnormalities and activation of inflammation-related gene expression. Behavioural characterization of the mutant line demonstrated stereotypic behaviours, hyperactivity, and deficits in social interactions with other fish. One key finding in the mutant line was the detection of reduced expression of a gene involved in the metabolism of melatonin. At least partly it was possible to rescue the behavioural deficits of *nomo1* mutants with the exogenous application of melatonin. These results highlight the usefulness of clinical application of melatonin in 16p13.11-associated neuropsychiatric disorders.

The authors use a battery of different techniques to collect their data. This is really the strong point of this study. The mutant is not only assessed in several behavioural assays (at different developmental stages) but also molecular analysis (transcriptome analysis, Realtime-PCR, LC-MS, HPLC) played a strong role in this study. To me, the logic behind each experiment is well explained and reasonable and there are no major flaws associated with the analysis or data presentation as far as I can see. However, before publication in *EMBO reports* I would recommend addressing the following points (minor revision needed) to improve the quality and readability of the manuscript:

A more general comment first: The manuscript suffers partly from language issues. Sometimes words in sentences are missing or grammar is not correct. As English is not my mother tongue, I don't feel qualified to make the proper corrections here, but at least somebody should proof read the manuscript who is better suited for the job than I am).

Specific things:

- Please indicate how many backcrosses were conducted before the molecular and behavioural analysis were carried out. Was the analysis done at the F1 level?
- Please indicate how the realtime PCR was done. How many technical replicates were used for each biological n? As you have used the $\Delta\Delta Ct$ method, what were your "housekeeping genes" to compare your gene of interest with? How many genes have you used as housekeepings? How to find genes being stable in expression throughout development (12hpf up to 2 mpf)? Have you used the same throughout development? Just curious.
- Sometimes the number of animals per group are reported in the text (rarely) sometimes in the figure legend (rarely) but mostly the information is missing. Why not including this info directly in the figure (write the number in the bar chart). This will help the reader to judge the variation and significance of the data presented.
- To better understand the impact of the 1-bp deletion you have introduced you should consider adding the amino acid sequence of *Nomo1* and show to what extent the protein sequence is lost due to the deletion. You could add the sequences in Figure 1b under the splicing scheme. By the way, what is the DUF2012 domain doing? There is no explanation in the text. Is it relevant for

its role as a nodal inhibitor? Is it a suppressor of inflammation?

- Are you sure that the HE stainings presented in 1D are "loosened and fragile" (line 306)? To me, it looks more like an artifact of the preparation procedure. How often did you observe this in each genotype?
- How was the data in Figure 2f calculated? Obviously, it was not, as indicated in the figure, L1-D1. And please change the color of the bars. You usually use black for +/+ and red for -/-. In Figure 2f the same colors mean different things. This is confusing and can be easily avoided just by using a different color here.
- In Figure 3. Please report the number of animals. Exchange the position in the panel of 3f and 3e. Would make more sense.
- In Figure 4, again, please report the number of animals.
- If there are space restrictions, I would recommend shifting Figure 5 to the supplement. It is interesting data, but just a side finding and doesn't bring the story forward. The next two figures are much more important.
- In Figure 6f, g, and h we have the same color-coding issue. Please use the colors you have introduced in 6c for f, g and h as well. Otherwise, it is too confusing.
- I can't find a reference in the result section to Figure 7c. It is only mentioned in the discussion. Please add a bit of description in the results section. Same color-coding issue in 7g, h and i.
- In relation to the wealth of data presented here in this manuscript, the discussion is rather short and superficial and needs some improvement. You could add for instance a section on the clinical implications of your findings. Would recommend using melatonin in 16p13.11 associated psychiatric disorders? Was this already tried? You could give an outlook what are the next steps to find the molecular switches regulating asmt? Is the promotor of asmt characterized? How could a nodal inhibitor act on this? Same for the induction of inflammation related genes. And so on.

Best wishes for this story

Referee #3:

In this manuscript, the authors showed that *nomo1* mutant zebrafish showed abnormal mid- and hindbrain development and exhibit multiple neuropsychiatric behaviors. While exogenous melatonin treatment may partially rescue the effect. This study suggested the potential of melatonin supplementation as a therapeutic regimen for neuropsychiatric disorders caused by *nomo1* deficiency, but some of the data is less convincing. The study reads more descriptive and lacks specific mechanisms in general. I have some major and minor questions.

Major points:

1. qPCR showed that neuro progenitor marker genes were down-regulated during early development. But SRM/MRM performed using the brains of juvenile zebrafish revealed a general increase of the level of neurotransmitters. The mechanism underlying the change of neurotransmitter level in *nomo1* mutant is unclear. The development of specific type of neuron should be analyzed at early stage, or the neural activity should be examined in the juvenile zebrafish to clarify whether the neural development or the neural activity was affected.
2. Related to above. Higher level of serotonin was detected by SRM/MRM and HPLC in *nomo1* mutant. Except for the abnormal serotonin metabolism, could this be resulted from change of 5-HT neuron activity or abnormal serotonin synthesis? Otherwise, do the expression level of genes involved in 5-HT neuron development, 5-HT synthesis, transport or reuptake changed?
3. Since serotonin can also function in anxiety and depression, whether reduction of serotonin level can rescue the abnormal behavior too?
4. In general, the characterization of several behaviors was relatively rough. Specific details of such parameters should be provided. For examples, the repetitive behaviors should be better described in terms of head and tail angles or even group activities, not just on the movement trajectory of a single fish.

Minor points:

1. Figure 1C: Have you checked the protein level? How to explain the reduction of mRNA level? Non-sense mediated decay?
2. Figure S2: What dose other developmental malformation refers to? It should be described in detail.
3. Figure S2: From S2C, it seems that the blood fluid was also affected, some mutants showed pericardium edema, dose *nomo1* also affected hematopoiesis?
4. Figure 1D: It seems that the telencephalon showed some difference, have you done the tissue section analysis of telencephalon? And besides the tissue analysis, which type of neuron was affected in *nomo1*-/-?
5. Figure 2: Since the locomotor activity was lower in *nomo1* mutant during the whole period, have you checked whether the locomotor ability was affected at early stage?
6. Figure 2G: What does the increasing of thigmotaxis reflect? Anxiety behavior? This should be discussed.
7. Figure 5B and Figure S3: How many fish were analyzed for WISH? Although no significant difference for forebrain markers was claimed based on the qPCR, but it showed a decrease trend, and it seems that the expression of *fezf2* and *th* were reduced from the image of WISH.

Referee #1:

In this study, Zhang et al. generated a *nomo1*^{-/-} zebrafish and examined the role of *nomo1* in neurodevelopment, neuroinflammation and behaviour. They found that the *nomo1* mutant zebrafish exhibit behavioural defects include autism-like repetitive behaviors and social defects. At the molecular level, they showed that *nomo1* deficiency affects neuronal development (downregulation of *huc*, *islet1*, *neurog1*), neurotransmitter metabolism and that melatonin treatment significantly rescued the certain behaviours of *nomo1*^{-/-} zebrafish.

Overall, this manuscript provides new insights into the role of *nomo1* in brain development and behaviour. The mechanistic links between *nomo1*, developmental defects, behavioural phenotypes and melatonin rescue remain somewhat weak. Experiments to strengthen these links in both larval and adult stages can significantly strengthen the mechanistic conclusions of the manuscript.

Major comments:

Line 299-300 "these morphological abnormalities of *nomo1*^{-/-} zebrafish gradually became less noticeable during development (Fig. S2C-E)" - Is it because those with abnormalities die prematurely? A survival curve can be helpful. Do homozygous *nomo1*^{-/-} survive to adult stage?

Response: The morphological abnormalities observed in *nomo1*^{-/-} zebrafish prior to 36 hpf were not lethal, as indicated by the survival curve presented in Fig EV1F. However, it is important to note that the developmental processes of mutant embryos are more vulnerable to external environmental factors. Consequently, during the early stages of development, *nomo1*^{-/-} zebrafish may exhibit transient and mild morphological abnormalities that are not directly attributed to the mutation itself. We discussed these results in revised manuscript at discussion section from 'we do observe' to 'mutation itself'.

Do *nomo*^{+/-} fish display any abnormalities?

Response: *nomo1*^{+/-} fish did not show any statistically significant morphological abnormalities, as depicted in Figure R1. Similar to the homozygous mutant, the *nomo1*^{+/-} fish exhibited no skeletal development abnormalities, as also shown in Figure R1. However, at 7 dpf, both the *nomo1*^{+/-} fish and the homozygous mutant displayed significantly reduced locomotor activity.

Figure for referee with unpublished data and its description has been removed upon request by the authors.

A low percentage of morphological abnormalities were observed in *nomo1*^{-/-} zebrafish according to the authors (Suppl. Figure 2). Representative figures in S2 do not show signs of bone defects - yet it is confusing that in the discussion the authors state at line 496-498 that locomotion defects in 7 dpf may be associated with impaired bone development in 7 dpf *nomo1*^{-/-} larvae. Were skeletal defects observed in *nomo1*^{-/-} zebrafish? Skeletal deformities can impact behaviour and melatonin plays a role in skeletal development- the rescue of behaviour may thus be misinterpreted. Additionally, Cao et al., 2018 (ref 11) showed that zebrafish *nomo* homozygous mutants exhibited chondrodysplasia.

Response: Thank you for your reminder. In the revised manuscript, we conducted Alcian Blue staining to assess cartilage and skeletal development, and no abnormalities were observed. The corresponding results are presented in Fig EV2H.

Regarding the *Nomo1* gene, it consists of a total of 31 exons. In this study, we identified that *Nomo1* transcription undergoes early termination at exon 7. In contrast, Cao et al. (2018) reported an early termination at exon 20 for *Nomo1*. Despite the truncation, the remaining approximately two-thirds of the amino acid chain, including the preserved EMC7-beta-sandw domain, suggests that the abnormally folded protein may still retain some functionality. However, this abnormal protein variant could potentially exert a different developmental impact compared to the normal *Nomo1* protein. We have revised the corresponding discussion section accordingly, from ‘as a result’ to ‘normal *Nomo1* protein’.

The *nomo1*^{-/-} brain in Fig1D seems to different in size. Important, the olfactory bulb and forebrain are clearly reduced in size in *nomo1*^{-/-} Fig 1D. Please provide the data for the brain size and a better representative brain for *nomo1*^{-/-} is required if there are no significant differences in *nomo1*^{-/-} and control brain size. Please also include section and staining of the forebrain in Fig. 1E.

Response: A better representative photograph depicting the brain of both *nomo1* ^{+/+} and *nomo1* ^{-/-} zebrafish is now presented in Figure 1D. More brain photos and size analysis of the forebrain, optic tectum, and cerebellum revealed no significant differences between the two brain types, are shown at appendix Fig S2. Furthermore, detailed sections of the forebrain and telencephalon are displayed in Fig 1E.

The authors observe a difference in the mass of the brain tissue in the model. Did the authors observe an increase in apoptosis? Are there specific regions of the brain affected in terms of reduced mass?

Response: In the revised manuscript, we conducted semi-quantitative analysis of apoptosis-related genes and neuronal marker expression in the adult zebrafish brain. We found that in the *nomo1*^{-/-} brain, there was an upregulation of the apoptosis-promoting gene *bbc3* and a downregulation of the apoptosis-inhibiting gene *bcl2l1* (Fig 5F). Additionally, the expression of various neuronal markers, such as *huc*, *neurog1*, *islet*, *egr2b*, and *foxb1a*, was significantly suppressed (Fig 5E).

Further examination of adult brain tissue through slicing revealed a reduction in the number of neurons in the forebrain VP, PM, and Pp nucleus, as well as the Ha nucleus in the telencephalon. Additionally, there was a decrease in neuronal cells in the Vam and VamG nucleus of the midbrain. These reductions in cell populations within this nucleus, coupled with the observed increase in brain cell apoptosis, likely contribute to the decline in brain quality associated with *nomo1* deficiency (Fig 1E).

Important markers are found down-regulated in 48 hpf *nomo1*^{-/-} fish such as *huc*, *neurog1*, *islet1* (Fig. 5). Are they still down-regulated in adult fish? Are there less neurons in larval brain? Brain sections could have been more thoroughly analysed at larval stages.

Response: In adult *nomo1*^{-/-} zebrafish, the expression of genes such as *huc*, *neurog1*, and *islet1* remained downregulated. Additionally, genes such as *egr2b*, *foxb1a*, *fezf*, *sim1a*, and *gad1b*, which did not show significant differences in expression at 48 hpf, were significantly suppressed. In the revised manuscript, we employed transgenic zebrafish lines expressing EGFP to label GABAergic inhibitory neurons, dsRed to label glutamatergic excitatory neurons, and RFP to label neuronal precursor cells. Through direct observation, we discovered a significant reduction in the numbers of various neuronal populations in the *nomo1*^{-/-} brain from 48 hpf throughout the larval stage (Fig 5).

Was there a difference in expression of the motor neuron marker *islet-1* in the spinal cord? Alternations in motor neuron number can also impact motor behaviour.

Response: At 30 hpf, WISH analysis of *islet1* indicated no significant differences in the number of motor neurons and sensory neurons (Fig R2). Furthermore, the numbers of GABAergic neurons, glutamatergic neurons, and neural progenitors were all observed to be decreased. This suggests that the decreased locomotor activity observed in *nomo1* zebrafish at 7 dpf is not solely due to the impairment of motor neurons but is instead a result of developmental abnormalities affecting multiple neuronal populations throughout the brain. We further discussed it at discussion section from 'brain functionality' to 'reduced locomotor activity'.

Figure for referee with unpublished data and its description has been removed upon request by the authors.

Figure legends do not state the statistical test performed for each analysis; these need to be stated. Particularly, in the methods (line 267), the authors state the groups were analysed using paired Student's t-test and such a statistical test is not appropriate for most datasets presented in this study. An unpaired t-test should be used when comparing WT vs mutant data for instance.

Response: Thanks a lot for your reminder. We do use unpaired t-test for comparing analysis in this manuscript. The accidentally wrong written methods were revised, and statistical test were stated for each figure legends.

The authors state that "repetitive stereotypic behaviors of small circle swimming were restored to normal

levels (Fig. 6H)" with MC treatment, but not the other repetitive behaviors. Any explanation for why there is a selective restoration in behavioural defects with MC treatment?

Response: According to Bodish et al.'s classification of repetitive stereotypic behaviors, where stereotypic behaviors are simpler and refer to apparently purposeless, unconscious movements such as, hand clapping, body shaking, arm waving, hand and finger movements, etc. We believe that small circle swimming may fall into this category. Another type of repetitive behavior, compulsive behavior, is more complex and refers to behaviors that are repeated or performed according to certain rules, e.g., arranging certain objects in a specific order, constantly checking doors, windows, drawers, counting, etc. We speculate that big circle swimming and back and forth may belong to this category. MC may restore simple stereotyped behaviors to normal levels by suppressing certain specific types of neuroinflammatory responses. The discussion was revised from 'there has been extensive research' to 'manifestations of neuropsychiatric disorders'.

Is melatonin decreased in *nomo1*^{-/-} fish? Can melatonin be evaluated as well by HPLC to support their claim?

Response: We detected melatonin level of zebrafish brain with LC-MS. Results were shown in Fig 7C.

The mechanistic link between *nomo1*, reduced expression of neuronal progenitor and mature neuron markers at 48 hpf, abnormal behavioural phenotypes and increase in inflammatory pathways, serotonin and ASMT/melatonin remain weak (including in the discussion). Can melatonin restore behavioural defects in 7 dpf larvae?

Response: The mechanistic link between *nomo1*, reduced expression of neuronal progenitor and mature neuron markers at 48 hpf, abnormal behavioral phenotypes and increase in inflammatory pathways, serotonin and ASMT/melatonin have been extensively discussed in the discussion section from 'Nodal proteins play' to 'melatonin supplementation facilitates the restoration of abnormal behavioral phenotypes'.

As illustrated in Fig R3, melatonin not only failed to restore the behavioral deficits in 7 dpf zebrafish larvae but also significantly suppressed the spontaneous locomotor activity in *nomo1*^{-/-} larvae. This could be attributed to the inherent sedative effects of melatonin. Despite adjusting the dosage based on larval weight and relevant literature, it is possible that the dosage of melatonin used in our study was relatively high. Furthermore, due to the limited complexity and motor capabilities of zebrafish larvae at 7 days old, there is no definitive consensus on whether they exhibit stable repetitive stereotypic behavior.

However, repetitive stereotypic behavior is considered a core behavioral phenotype in individuals with autism spectrum disorder. Therefore, the results of melatonin improving repetitive stereotypic behavior in adult fish, as presented in this manuscript, hold greater clinical significance and research value in the context of autism etiology research.

Figure for referee with unpublished data and its description has been removed upon request by the authors.

How can defects in social behaviour be explained upon *nomo1* deficiency? Neuronal network in *nomo1* brain at larval and adult stages can be thoroughly examined by immunostaining.

Response: Analysis of sections of adult zebrafish brain showed a significant reduction in cells in the forebrain Vp nucleus, which is homologous to the mammalian amygdala, and previous studies have found that social deficits are associated with abnormalities in the amygdala, so the reduction in neurons in this nucleus may be associated with social deficits. However, this is only a partially explainable cause. The loss of *Nomo1* results in developmental abnormalities in various neuronal populations, including neuronal precursors and motor neurons, during early embryonic development. As development progresses, the progressive loss of *Nomo1* adversely affects an increasing number of neuronal populations and neurotransmitters, thus there should be multiple nucleus, multiple neurons, and multiple neurotransmitter abnormalities combined to cause social deficits.

Thank you very much for your valuable suggestion. You have raised a valid point regarding the identification of the specific neural activity or neural networks impacted by the *nomo1* mutation, highlighting its significance. While we acknowledge the importance of such investigations, we would like to emphasize that our manuscript primarily focuses on the pressing need to explore the etiology and intervention strategies targeting these domino-like mechanisms.

The dysfunction of *nomo1*, a gene with diverse functional implications, is likely mediated through an inflammatory response, which subsequently affects multiple neural circuits and gives rise to various phenotypes. Both intervention strategies targeting inflammatory responses and studying the impact on neural circuits are worthwhile avenues of research.

Minor comments:

In the introduction, the authors mention the role of *NOMO1* in the inhibition of the Nodal pathway. What is the significance of this role in the study? Were the genes in the Nodal pathway perturbed in the transcriptomic analyses? This should be commented.

Response: Nodal proteins play a crucial role as inducers of mesendoderm formation, but their signaling must be blocked in the prospective neuroectoderm for proper nervous system development. *Nomo1*, along with the nodal signaling antagonists *nicalin* and *TMEM147*, forms a complex that effectively inhibits Nodal signaling. Deletion of *nomo1* disrupts the blocking effect of Nodal signaling in the ectoderm, resulting in widespread abnormalities in nervous system development. Transcriptome analysis further revealed that, apart from *Nomo1*, the expression of Nodal signaling-related genes remained unaltered (Table below presents the GO enrichment analysis results for the Nodal signaling

pathway, with only Nomo1 being affected in *nomo1*^{-/-} zebrafish). The manuscript was revised from 'Nodal proteins play' to 'signaling in the mesendoderm'.

GO description	Gene Ratio	P value	P adj	Gene ID	Gene name
Nodal signaling pathway	1/259	0.17038501	0.77744578	ENSDARG0000078592	Nomo1

At what age (be more specific than juvenile) was transcriptomic analysis performed?

Response: The transcriptomic analysis was performed at 2mpf, manuscript was revised at line 207.

Only dpf is defined in the method section, please define hpf and mpf as well.

Response: The manuscript was revised at Line 139 and Line 157.

Genes should be italics (line 273)

Response: Thanks for your reminder, manuscript was revised.

Please use "did not" instead of "didn't"

Response: The manuscript was revised.

7dpf are larvae not embryo (line 278, line 496)

Response: Thanks for your reminder, the manuscript was revised.

Graphs could be better presented as dot plots, with the average and appropriate error bars indicated

Response: Thank you for your constructive advice. In revised manuscript, graphs are presented as dot plots, with the average shown as column and the error bars indicating SEM.

Line 321, "confirming the specificity of the phenotype" - What is meant by this strong statement? What specific phenotype is the authors referring to here?

Response: The specific phenotype was affected locomotor activity of *nomo1*^{-/-}. Thank you for the reminder, the manuscript has been properly revised to avoid misunderstanding.

The Minocycline-treated animals were kept in a light-depleted environment. Could this have an influence on motor behaviour? Line258

Response: Sorry for the misunderstanding caused by the lack of precision in the original manuscript. Generally, the surface light intensity of the water column at our lab are about 300 lux, while the surface light intensity of the minocycline-treated fish and its control group are adjusted to about 200 lux to prevent minocycline from being fully decomposed. Both light intensities were under the normal living light conditions of zebrafish and did not affect their motility. The manuscript was revised at Line 276.

Explaining the thigmotaxis test and its choice - could be beneficial in understanding the clinical relevance of the read-out.

Response: Besides locomotor activity, thigmotaxis is another validated index of anxiety. Animals that are engaged in thigmotaxic behavior tend to move in close proximity to the boundaries of the environment. This explanation was added to the manuscript at Line 403-Line 405.

Disparities in the colour of the graphs can be a bit confusing (for example: Figure 4D, E, G, I; Figure 7.H)

Response: Thanks for your reminder, the color of the graphs was revised.

Figure 3B, statistics should be included.

Response: Figures 3B, D, and E illustrate the distribution of movement distances per minute for different genotypes of zebrafish over a 30-minute period during the complete experiment. The statistical analysis of this data is presented in Figure 3F, which depicts the average movement distances per minute for different genotypes of zebrafish.

Referee #2:

Dear authors, co-reviewers and editors,

the manuscript entitled “NOMO1 deficiency results in abnormal behavior and disturbed serotonin metabolism, melatonin supplementation exhibited rescue effect” by Zhang and Li et al., describes the generation and analysis of a *nomo1* deficient zebrafish line. The motivation to analyze *nomo1* is based on the chromosomal localization of the human gene in 16p13.11, a region which is involved in several neuropsychiatric disorders. The authors demonstrate that loss of *nomo1* results in mild morphological abnormalities and activation of inflammation-related gene expression. Behavioural characterization of the mutant line demonstrated stereotypic behaviours, hyperactivity, and deficits in social interactions with other fish. One key finding in the mutant line was the detection of reduced expression of a gene involved in the metabolism of melatonin. At least partly it was possible to rescue the behavioural deficits of *nomo1* mutants with the exogenous application of melatonin. These results highlight the usefulness of clinical application of melatonin in 16p13.11-associated neuropsychiatric disorders.

The authors use a battery of different techniques to collect their data. This is really the strong point of this study. The mutant is not only assessed in several behavioural assays (at different developmental stages) but also molecular analysis (transcriptome analysis, Realtime-PCR, LC-MS, HPLC) played a strong role in this study. To me, the logic behind each experiment is well explained and reasonable and there are no major flaws associated with the analysis or data presentation as far as I can see. However, before publication in EMBOreports I would recommend addressing the following points (minor revision needed) to improve the quality and readability of the manuscript:

A more general comment first: The manuscript suffers partly from language issues. Sometimes words in sentences are missing or grammar is not correct. As English is not my mother tongue, I don't feel qualified to make the proper corrections here, but at least somebody should proof read the manuscript who is better suited for the job than I am).

Responses: Thank you, we have thoroughly refined the English composition, grammar, and sentence structure to improve the overall fluency and comprehension.

Specific things:

- Please indicate how many backcrosses were conducted before the molecular and behavioural analysis were carried out. Was the analysis done at the F1 level?

Response: The *Nomo1* homozygous deletion mutation in zebrafish was generated in 2014 and had been backcrossed over 6 times by the moment of experiment.

- Please indicate how the realtime PCR was done. How many technical replicates were used for each biological n? As you have used the $\Delta\Delta Ct$ method, what were your "housekeeping genes" to compare your gene of interest with? How many genes have you used as housekeepings? How to find genes being stable in expression throughout development (12hpf up to 2 mpf)? Have you used the same throughout development? Just curious.

Response: Each biological n has four technical replicates. According to Tang et al¹, The β -actin gene is stable in expression throughout development, for its fundamental role in cell motility. Therefore, we use β -actin as housekeeping genes. The methods section of realtime PCR were revised at Line 130-Line 131.

- Sometimes the number of animals per group are reported in the text (rarely) sometimes in the figure legend (rarely) but mostly the information is missing. Why not including this info directly in the figure (write the number in the bar chart). This will help the reader to judge the variation and significance of the data presented.

Response: Thanks for your reminder. We added number of animals in the figure legend and in revised manuscript, graphs are presented as dot plots to show the sample size more visually, with the average shown as column and the error bars indicating SEM.

- To better understand the impact of the 1-bp deletion you have introduced you should consider adding the amino acid sequence of Nomo1 and show to what extent the protein sequence is lost due to the deletion. You could add the sequences in Figure 1b under the splicing scheme. By the way, what is the DUF2012 domain doing? There is no explanation in the text. Is it relevant for its role as a nodal inhibitor? Is it a suppressor of inflammation?

Response: Thanks for your reminder, the amino acid sequence of Nomo1 is shown in Figure EV1. Wild type Nomo1 consists of 1220 amino acids, while mutant Nomo1 only consists of 232 amino acids. According to pfam database, DUF means domain of unknown function. Recently, DUF2012 was renamed to EMC7-beta-sandw (PF09430) in the updated description of this domain in the pfam database. EMC7-beta-sandw is a beta-sandwich domain found in ER membrane protein complex subunit 7(EMC7), which is an integral membrane component of the EMC². EMC mediate the insertion of newly synthesized membrane proteins into endoplasmic reticulum membranes³. EMC7-beta-sandw domain may be associated with the embedding of Nomo1 in the ER membrane. Figure1 and the manuscript was revised.

- Are you sure that the HE stainings presented in 1D are "loosened and fragile" (line 306)? To me, it looks more like an artifact of the preparation procedure. How often did you observe this in each genotype?

Response: The experimental procedure for brain slicing and HE staining was the same for both wild-type and mutant samples. However, HE staining of the three mutant brains' slices consistently revealed the presence of fissures, whereas such occurrences were minimal in the wild-type (figure below). Given the reduced cell count and increased apoptosis (Fig 5), we speculate that the tissue fragility observed in *nomo1*^{-/-} brains is a result of the combined effects of these factors.

Figure for referee with unpublished data and its description has been removed upon request by the authors.

- Figure 2D: a single example is not sufficient to draw the conclusions. The result needs to either be somehow quantified, or a supplementary figure must be added that shows multiple independent examples.

Response: Thank you for your reminder. In the revised manuscript, we present multiple independent brain samples in appendix Figure S2. We also conducted quantitative analysis of the size of the forebrain, optic tectum, and cerebellum regions. The results revealed no significant differences in size between the wild-type and mutant brains.

- How was the data in Figure 2f calculated? Obviously, it was not, as indicated in the figure, L1-D1. And please change the color of the bars. You usually use black for +/+ and red for -/-. In Figure 2f the same colors mean different things. This is confusing and can be easily avoided just by using a different color here.

Response: Fig. 2F showed the ratio of the average distance in dark to the average distance in light. The manuscript was revised at Line 392. To avoid misunderstanding, we change the label and the color of Fig. 2F.

- In Figure 3. Please report the number of animals. Exchange the position in the panel of 3f and 3e. Would make more sense.

Response: Fig3 and legend was revised as suggested.

- In Figure 4, again, please report the number of animals.

Response: Fig4 and legend was revised as suggested.

- If there are space restrictions, I would recommend shifting Figure 5 to the supplement. It is interesting data, but just a side finding and doesn't bring the story forward. The next two figures are much more important.

Response: The original Figure 5 was shifted to Fig EV3.

- In Figure 6f, g, and h we have the same color-coding issue. Please use the colors you have introduced in 6c for f, g and h as well. Otherwise, it is too confusing.

Response: Fig6 and legend was revised.

- I can't find a reference in the result section to Figure 7c. It is only mentioned in the discussion. Please add a bit of description in the results section. Same color-coding issue in 7g, h and i.

Response: The original Figure 7c was shifted to appendix Fig S8.

- In relation to the wealth of data presented here in this manuscript, the discussion is rather short and superficial and needs some improvement. You could add for instance a section on the clinical implications of your findings. Would recommend using melatonin in 16p13.11 associated psychiatric disorders? Was this already tried? You could give an outlook what are the next steps to find the molecular switches regulating asmt? Is the promotor of asmt characterized? How could a nodal inhibitor act on this? Same for the induction of inflammation related genes. And so on.

Response: Thank you a lot for your reminder. We rewrite the discussion section based on the current results.

Reference:

1. Tang, R., Dodd, A., Lai, D., McNabb, W. C. & Love, D. R. Validation of zebrafish (*Danio rerio*) reference genes for quantitative real-time RT-PCR normalization. *Acta Biochim. Biophys. Sin. (Shanghai)*. **39**, 384–390 (2007).
2. Pleiner, T. *et al.* Structural basis for membrane insertion by the human ER membrane protein complex. *Science* **369**, 433–436 (2020).
3. O'Donnell, J. P. *et al.* The architecture of EMC reveals a path for membrane protein insertion. *Elife* **9**, (2020).

Referee #3:

In this manuscript, the authors showed that *nomo1* mutant zebrafish showed abnormal mid- and hindbrain development and exhibit multiple neuropsychiatric behaviors. While exogenous melatonin treatment may partially rescue the effect. This study suggested the potential of melatonin supplementation as a therapeutic regimen for neuropsychiatric disorders caused by *nomo1* deficiency, but some of the data is less convincing. The study reads more descriptive and lacks specific mechanisms in general. I have some major and minor questions.

Major points:

1. qPCR showed that neuro progenitor marker genes were down-regulated during early development. But SRM/MRM performed using the brains of juvenile zebrafish revealed a general increase of the level of neurotransmitters. The mechanism underlying the change of neurotransmitter level in *nomo1* mutant is unclear. The development of specific type of neuron should be analyzed at early stage, or the neural activity should be examined in the juvenile zebrafish to clarify whether the neural development or the neural activity was affected.

Response: In revised manuscript, we conducted further analysis of development of multiple neurons (Fig 5). Results showed that at early development stage, *huc* indicated neuro progenitors, *islet* indicated motor neuron, *egr2b* indicated hindbrain neuron and *foxb1a* indicated ectoderm neuron were all inhibited (Fig 5A-D, Fig EV3). And as development proceeds, not only fore-mentioned neurons, the former un-affected *sim1a* indicated dopaminergic neurons, *fezf2* indicated forebrain neurons, *gad1b* indicated GABAergic neuron and *vglut2a* indicated glutamatergic excitatory neurons were all inhibited (Fig 5E). At adult stage, brain of *nomo1*^{-/-} exhibited fragile tissues, less mass and up-regulated apoptosis and neuroinflammation signaling pathways (Fig 1D-E, Fig 5F, Fig 6A-B). These results indicate that the deficiency of *nomo1*, a regulatory protein involved in the widely expressed nodal signaling pathway, can give rise to various neuronal developmental abnormalities during early stages. These abnormalities progressively worsen as development proceeds, thereby triggering a cascade of stress responses and neuroinflammation within the surrounding environments. We discussed it in discussion section from ‘Nodal proteins play’ to ‘behavior of small circling’.

Thank you so much for your valuable suggestion of identify the specific neural activity or neural networks impacted by the *nomo1* mutation, highlighting its significance. While we acknowledge the importance of such investigations, however, *nomo1* deficiency leads to multiple neuronal abnormalities, and the neural circuits involved are complex and difficult to distinguish. And we would like to emphasize that our manuscript primarily focuses on the pressing need to explore the etiology and intervention strategies targeting these domino-like mechanisms.

The dysfunction of *nomo1*, a gene with diverse functional implications, is likely mediated through an inflammatory response, which subsequently affects multiple neural circuits and gives rise to various phenotypes. Both intervention strategies targeting inflammatory responses and studying the impact on neural circuits are worthwhile avenues of research.

2. Related to above. Higher level of serotonin was detected by SRM/MRM and HPLC in *nomo1* mutant. Except for the abnormal serotonin metabolism, could this be resulted from change of 5-HT neuron activity or abnormal serotonin synthesis? Otherwise, do the expression level of genes involved in 5-HT neuron development, 5-HT synthesis, transport or reuptake changed?

Response: Thanks for your reminder. Transcriptome analysis showed *Nomo1* deficiency didn't affect expression level of genes involved in 5-HT neuron development, 5-HT synthesis, transport or reuptake process including *tph2*, *slc7a5*, *ddc*, *slc18a2* and so on (Table EV4).

3. Since serotonin can also function in anxiety and depression, whether reduction of serotonin level can rescue the abnormal behavior too?

Response: Thanks for your reminder, we treated zebrafish with p-chlorophenylalanine (pCPA), a 5-HT inhibitor, to investigate whether the abnormal behavior could be rescued. The results showed that pCPA treated zebrafish showed rescued locomotor activity. Bad news is that treated zebrafish showed an increase in inter-individual distance in the shoaling experiment compared to untreated zebrafish (Fig. EV4 A). In contrast, pCPA treated zebrafish showed no significant differences in social preference and repeated stereotyped behavior analyses. This suggests that the genesis of anxiety-like behavior in *nomo1*^{-/-} zebrafish is also related to elevated 5-HT levels. We discussed it at discussion section from ‘increased serotonin levels’ to ‘treating autism’.

4. In general, the characterization of several behaviors was relatively rough. Specific details of such parameters should be provided. For examples, the repetitive behaviors should be better described in terms of head and tail angles or even group activities, not just on the movement trajectory of a single fish.

Response: The analysis of repetitive behaviors was achieved with Viewpoint, a commonly used software for professional analysis of zebrafish behavior produced by a French manufacturer. After videotaping the zebrafish movements, the software automatically analyzes the coordinates of the zebrafish's position at each moment and the current degree of movement (defined by viewpoint as the position of 1,000+ points mixed with color shades). The data were obtained by filtering out videos with specific variations in motility and coordinates, such as back and forth swim, big circling swim, and small circling swim, and then we manually validating them. This analysis process does not involve head and tail angles, as shown in the video EV5-7; and the movement trajectory of each fish is analyzed individually, it doesn't involve the group.

Minor points:

1. Figure 1C: Have you checked the protein level? How to explain the reduction of mRNA level? Non-sense mediated decay?

Response: A WB test would be ideal to demonstrate functional reduction of *Nomo1*. Unfortunately, however, biologics companies such as Santa Cruz Biotechnology, NOVUS Biologicals, Biocompare, GeneTex, Abcam, and Abmart do not offer commercially available antibodies to *Nomo1* for sale. In the present manuscript, qRT-pcr results showed a significant reduction in *Nomo1* mRNA and our experimental results were quite specific and significant, so we concluded that its highly unlikely *Nomo1* was not knocked out. And I think you made an excellent point, the reduction of mRNA level might be induced by non-sense mediated decay.

2. Figure S2: What dose other developmental malformation refers to? It should be described in detail.

Response: other developmental malformation indicate the malformation of yolk, including smaller yolk, bigger yolk and yolk extension malformation. The manuscript was revised at the result section ‘Morphological analysis of *nomo1*^{-/-} zebrafish revealed abnormalities in early development and adult brain’.

3. Figure S2: From S2C, it seems that the blood fluid was also affected, some mutants showed pericardium edema, dose *nomo1* also affected hematopoiesis?

Response: Nomo1 deficiency didn't affect hematopoiesis of zebrafish, Transcriptome analysis shows hematopoiesis-related genes expression was unaffected (see table below). Some mutants exhibit pericardial edema, which can be attributed to the environmental influences. Fish with genetic defects may be more fragile and their developmental processes are more susceptible to external environmental influences, thus showing temporary morphological abnormalities early in embryonic development. And these abnormalities can be corrected as development proceeds. We discussed these results in revised manuscript at discussion section from 'we do observe' to 'mutation itself'.

gene_name	TU_fpk	NOMO_fpk	NOMOvsTU_log2Fold Change	NOMOvsTU_padj
tal1	20.7590889	21.2909199	0.04540038	1
lmo2	14.8244031	16.9857918	0.20521027	1
gata2a	5.76201416	5.83766125	0.02771632	1
fli1a	1.93522902	2.08773639	0.11820401	1
fli1b	1.46713417	1.59305786	0.127491	1
etv2	0.5068548	1.08542827	1.09940506	1
kdr1	8.3764599	7.40788169	-0.1683397	1
gata1a	0.23424301	0.60195698	1.35666183	0.97500541
ptpn6	1.8895537	2.04873556	0.12539075	1
MFAP4	0.55361042	2.60144705	2.22085123	0.09543091
runx1	0.9504462	0.55827514	-0.7410216	1
gata2b	0.09417647	0.03226859	-1.42897	1

4. Figure 1D: It seems that the telencephalon showed some difference, have you done the tissue section analysis of telencephalon? And besides the tissue analysis, which type of neuron was affected in *nomo1*^{-/-}?

Response: There is no difference in telencephalic size between *nomo1*^{+/+} and *nomo1*^{-/-}, and the difference shown in the original image is due to the angle at which the photo was taken. We replaced a figure 1D that would not cause misunderstanding, and added multiple brain morphology maps, as well as graphs for statistical analysis of brain size (appendix Fig S2).

During zebrafish development, a variety of neurons are affected, including the *sim1a* indicated dopaminergic neurons, *fezf2* indicated forebrain neurons, *gad1b* indicated GABAergic neuron and *vglut2a* indicated glutamatergic excitatory neurons and so on (Fig 5E).

5. Figure 2: Since the locomotor activity was lower in *nomo1* mutant during the whole period, have you checked whether the locomotor ability was affected at early stage?

Response: At 48hpf, motoneuron marker expression was decreased in *nomo1* deficiency zebrafish, and at 72hpf, the number of excitatory neuron cells was reduced. the decreased locomotor activity performance in zebrafish at 7dpf may be related to the down-regulation of the expression of these two types of neurons. However, after 15dpf, the locomotor activity performance of zebrafish was up-regulated until 3mpf, which suggests that the locomotor ability of zebrafish was not affected after 15dpf. We further discussed it at discussion section from 'brain functionality' to 'reduced locomotor activity'

6. Figure 2G: What does the increasing of thigmotaxis reflect? Anxiety behavior? This should be discussed.

Besides locomotor activity, thigmotaxis is another validated index of anxiety. Animals that are engaged in thigmotaxic behavior tend to move in close proximity to the boundaries of the environment. This explanation was added to the manuscript at Line 403- Line 405.

7. Figure 5B and Figure S3: How many fish were analyzed for WISH? Although no significant difference for forebrain markers was claimed based on the qPCR, but it showed a decrease trend, and it seems that the expression of fezf2 and th were reduced from the image of WISH.

Response: WISH was done in two replicates with 10 embryos each. You are right, indeed the qPCR all show a decreasing trend, and the WISH results for fezf2 and th do seem to show a decrease. Considering that the area of decreased signal exhibited in the WISH results is small and does not account for a large percentage of the total signal, the qPCR results did not show a statistical difference, and we consider the results are reasonable.

Dear Prof. LI

Thank you for the submission of your revised manuscript to EMBO reports. Since my colleague Esther Schnapp is currently traveling, I have temporarily taken over the handling of your manuscript. We have now received the full set of referee reports that is copied below.

As you will see, all referees are very positive about the study and request only minor changes, which should be addressed in this final round of revision.

From the editorial side, there are also a few things that we need before we can proceed with the official acceptance of your study.

- Please update the references to the alphabetical Harvard style. The abbreviation 'et al' should be used if more than 10 authors. You can download the respective EndNote file from our Guide to Authors https://endnote.com/style_download/embo-reports/

- Please update the 'Conflict of interest' paragraph to our new 'Disclosure and competing interests statement'. For more information see <https://www.embopress.org/page/journal/14693178/authorguide#conflictsofinterest>

- Figure callouts: Fig. 1 should be called out before Fig. 2; missing callouts for: Fig. 3E, 7F, Table EV1, Table EV4; there are callouts for Fig. 6G and Fig. 7G-I, but no such panels

- Appendix: Please provide a title page with a table of content and page numbers. The nomenclature should be Appendix Figure S1-S8 with the appropriate callouts in the text.

- Appendix Figure legends - please correct/add the following:

Fig. S2A needs scale bars.

Fig. S2B lacks information of 'n' in the legend and a definition of whether x number of fish has been analyzed in one experiment (N) or in several repeats

Fig S6, S7, S8 lack a description of 'n'

Fig S4, S6, S7, and S8 lack a definition of "****" or "****"

In general: please replace "didn't" by "did not"

- We need the movies in the following format: Each movie is zipped with its legend (README.txt file) into one ZIP file. Then each ZIP file is uploaded individually.

- Tables EV1-EV4 and their legends should be removed from the zip file, and uploaded individually. The legends should be included in the corresponding Excel file.

- The manuscript sections are in the wrong order. Please order them like this:

Title page - Abstract - Introduction - Results - Discussion - Materials and Methods - Acknowledgements - Disclosure and competing interests statement - References - Figure legends - Tables and their legends (not EV tables) - Expanded View Figure legends

- I attach to this email a related manuscript file with comments by our data editors. Please address all comments and upload a revised file with tracked changes with your final manuscript submission. I have also suggested changes to the title and abstract - please review these as well.

- Finally, EMBO Reports papers are accompanied online by A) a short (1-2 sentences) summary of the findings and their significance, B) 2-3 bullet points highlighting key results and C) a synopsis image that is 550x300-600 pixels large (width x height) in PNG for JPG format. You can either show a model or key data in the synopsis image. Please note that the size is rather small and that text needs to be readable at the final size. Please send us this information along with the revised manuscript.

- On a different note, I would like to alert you that EMBO Press offers a new format for a video-synopsis of work published with us, which essentially is a short, author-generated film explaining the core findings in hand drawings, and, as we believe, can be very useful to increase visibility of the work. This has proven to offer a nice opportunity for exposure i.p. for the first author(s) of the study. Please see the following link for representative examples and their integration into the article web page:

<https://www.embopress.org/doi/full/10.15252/emj.2019103932>

With kind regards,

Referee #1:

The authors have made every effort to address all my comments on the original manuscript. The study remains more descriptive and lacks specific mechanisms, particularly on the connections between abnormal behavioural phenotypes and increase in inflammatory pathways, serotonin and ASMT/melatonin. However, I am particularly happy that they managed to significantly improve the manuscript with additional experiments and provide a better discussion for potential mechanisms involved.

I have minor comments:

For results in Figure 1E and 5, the authors conclude a significant reduction in the various neuronal population in the *nomo1*^{-/-}, can these be quantified?

As a respond to a previous comment about skeletal defects in *nomo1*^{-/-}: "In the revised manuscript, we conducted Alcian Blue staining to assess cartilage and skeletal development, and no abnormalities were observed. The corresponding results are presented in Fig EV2H." The reviewer appreciates the attempt to address the skeletal defects in *nomo1*^{-/-}. However, Alcian blue only marks cartilage and a bone staining, with Alizarin Red would have been provided more information on skeletal defects in the *nomo1*^{-/-} mutants.

Line 116-119, transgenic names should be in italics.

Referee #2:

All my points/concerns have been adequately addressed. Good luck with the publication. Nice paper.

Referee #3:

The authors now provided significant data to answer my question. I have no further questions.

Referee #1:

The authors have made every effort to address all my comments on the original manuscript. The study remains more descriptive and lacks specific mechanisms, particularly on the connections between abnormal behavioural phenotypes and increase in inflammatory pathways, serotonin and ASMT/melatonin. However, I am particularly happy that they managed to significantly improve the manuscript with additional experiments and provide a better discussion for potential mechanisms involved.

I have minor comments:

For results in Figure 1E and 5, the authors conclude a significant reduction in the various neuronal population in the *nomo1*^{-/-}, can these be quantified?

Response: Results in Figure 1E (now Figure 1F) and 5A-D were quantified and shown in Figure 1E, Figure 5E-F, respectively. Quantification methods were updated in “Preparation of paraffinized sections and HE staining” and “Fluorescence signal observation and quantification” section.

As a respond to a previous comment about skeletal defects in *nomo1*^{-/-}: "In the revised manuscript, we conducted Alcian Blue staining to assess cartilage and skeletal development, and no abnormalities were observed. The corresponding results are presented in Fig EV2H." The reviewer appreciates the attempt to address the skeletal defects in *nomo1*^{-/-}. However, Alcian blue only marks cartilage and a bone staining, with Alizarin Red would have been provided more information on skeletal defects in the *nomo1*^{-/-} mutants.

Response: The manuscript was revised as suggested at Fig EV2H.

Line 116-119, transgenic names should be in italics.

Response: Thank you for your reminder, the manuscript was revised in “Generation of transgenic line” section.

Referee #2:

All my points/concerns have been adequately addressed. Good luck with the publication. Nice paper.

Response: Thank you for your positive comment!

Referee #3:

The authors now provided significant data to answer my question. I have no further questions.

Response: Thank you for your positive comment!

Prof. Qiang LI
FUDAN UNIVERSITY
cardiovascular department
339 Wanyuan Road Minhang District Shanghai
orcid||||| 201102
China

Dear Prof. LI,

I am very pleased to accept your manuscript for publication in the next available issue of EMBO reports. Thank you for your contribution to our journal.

I would like to suggest one more change to the abstract, that needs to be written in present tense. Please let me know whether you agree with this:

"The Habenular nucleus and the pineal gland in the telencephalon are affected, and the melatonin level of *nomo1*^{-/-} is reduced."

Yours sincerely,
